# Adaptive Inference-Time Scaling via Cyclic Diffusion Search

**Gyubin Lee**[*]
KAIST
dlrbqls0521@kaist.ac.kr

**Truong Nhat Nguyen Bao**[*]
KAIST
truongnhatnguyenbao@kaist.ac.kr

**Jaesik Yoon**
KAIST & SAP
jaesik.yoon@kaist.ac.kr

**Dongwoo Lee**
KAIST
dongwoolee@kaist.ac.kr

**Minsu Kim**
Mila – Quebec AI Institute
KAIST
minsu.kim@mila.quebec

**Yoshua Bengio**
Mila – Quebec AI Institute
Université de Montréal
yoshua.bengio@mila.quebec

**Sungjin Ahn**
KAIST & NYU
sungjin.ahn@kaist.ac.kr

## Abstract

Diffusion models have demonstrated strong generative capabilities across domains ranging from image synthesis to complex reasoning tasks. However, most inference-time scaling methods rely on fixed denoising schedules, limiting their ability to allocate computation based on instance difficulty or task-specific demands adaptively. We introduce the challenge of adaptive inference-time scaling—dynamically adjusting computational effort during inference—and propose Adaptive Bi-directional Cyclic Diffusion (ABCD), a flexible, search-based inference framework. ABCD refines outputs through bi-directional diffusion cycles while adaptively controlling exploration depth and termination. It comprises three components: Cyclic Diffusion Search, Automatic Exploration-Exploitation Balancing, and Adaptive Thinking Time. Experiments show that ABCD improves performance across diverse tasks while maintaining computational efficiency.

## 1 Introduction

Diffusion models have become a leading class of generative models, achieving state-of-the-art performance across diverse domains, from high-fidelity image synthesis to complex language generation [21]. A core strength of diffusion models is their ability to represent complex, multimodal distributions through a hierarchical, multi-step denoising process [9, 27]. This iterative refinement makes them well-suited not only for content generation but also for challenging search and reasoning tasks, such as planning controls [1, 4], maze navigation [34, 35], and puzzle solving [7, 8].

Despite their success, a key open challenge is how to effectively realize *inference-time scaling*—the ability to improve model performance by allocating additional computation during inference, either on a per-task basis or, ideally, adaptively per instance. While several approaches have recently explored inference-time scaling for diffusion models [15, 16, 17], they still face notable limitations.

---

[*]Equal contribution. Correspondence to Gyubin Lee and Sungjin Ahn <dlrbqls0521@kaist.ac.kr and sungjin.ahn@kaist.ac.kr>.

Among the limitations, the most critical is the prevalent reliance on fixed inference procedures. Most existing methods—such as Best-of-N [36], Sequential Monte Carlo [24, 31], Beam Search [17], and Search-over-Path [16]—execute a predetermined number of denoising steps along a trajectory. Once this trajectory completes, the inference is terminated, regardless of the complexity or specific requirements of the input instance. This rigidity fundamentally restricts the model's ability to adapt computation based on instance difficulty, ambiguity, or the need for higher confidence or output quality. As a result, performance could suffer on more challenging inputs, and computation may be either underutilized or inefficiently spent.

In this paper, we propose a novel method—Adaptive Bi-directional Cyclic Diffusion (ABCD)—to address the challenge of *adaptive* inference-time scaling. This approach reframes diffusion model inference as a *flexible* and *efficient* search process, enabling adaptive computation and instance-aware refinement based on task difficulty and evolving solution quality. ABCD consists of three components. *Cyclic Diffusion Search* (CDS) enables iterative refinement by cycling bi-directionally through the diffusion process—alternating between denoising and re-noising steps—allowing the model to escape local minima and explore alternative generative trajectories. *Automatic Exploration-Exploitation Balancing* (AEEB) introduces a mechanism for adaptively controlling the depth of exploration by distributing particles across multiple re-noising levels, allowing the model to implicitly identify the appropriate amount of computation for each instance. Finally, *Adaptive Thinking Time* (ATT) provides a principled stopping criterion by monitoring the evolution of solution quality over inference cycles, ensuring computation is allocated efficiently and terminated when additional refinement yields diminishing returns. We demonstrate the effectiveness of ABCD across a range of tasks—including control planning, maze solving, Sudoku, and molecule generation—showing significant gains in inference-time flexibility, computational efficiency, and solution quality.

Our main contributions of this paper are as follows: (1) We formalize the problem of adaptive inference-time scaling in diffusion models and identify the limitations of fixed-step inference. (2) We propose Adaptive Bi-directional Cyclic Diffusion, a novel framework that frames inference as a flexible, search-based process with dynamic compute allocation. (3) We introduce three components—Cyclic Diffusion Search, Automatic Exploration-Exploitation Balance, and Adaptive Thinking Time—that enable iterative refinement, adaptive search control, and automatic stopping. (4) We empirically validate ABCD across diverse tasks, including planning, maze solving, Sudoku, and molecule generation, showing improvements in flexibility, accuracy, and compute efficiency.

## 2 Preliminaries

**Diffusion model.** Diffusion models [9, 25, 26, 27] have recently emerged as a powerful generative framework that formulates sample generation as a gradual denoising process. This framework consists of two complementary components: a fixed forward process, which incrementally injects noise into data to transform structured inputs into nearly pure noise, and a learned reverse process, which aims to recover clean data from noisy observations. Formally, the forward process defines a sequence of latent variables $\mathbf{x}_1, \mathbf{x}_2, \ldots, \mathbf{x}_T$ conditioned on the original sample $\mathbf{x}_0$ via a Markov chain: $q(\mathbf{x}_{t+1}|\mathbf{x}_t) = \mathcal{N}(\mathbf{x}_{t+1}; \sqrt{\alpha_t}\mathbf{x}_t, (1-\alpha_t)\mathbf{I})$, where $\alpha_t \in (0,1)$ denotes a pre-defined noise schedule. This process ultimately transforms $\mathbf{x}_0$ into Gaussian noise $x_T \sim \mathcal{N}(0, I)$. The forward transition at an arbitrary step $t$ is given by:

$$\mathbf{x}_t = \sqrt{\bar{\alpha}_t}\mathbf{x}_0 + \sqrt{1 - \bar{\alpha}_t}\epsilon, \quad \epsilon \sim \mathcal{N}(0, \mathbf{I}),$$

where $\bar{\alpha}_t = \prod_{s=1}^t \alpha_s$ denotes the cumulative product of the noise schedule. This expression provides the marginal distribution of $\mathbf{x}_t$ conditioned on $\mathbf{x}_0$.

The reverse process uses a neural network to predict the noise component $\epsilon_\theta(\mathbf{x}_t, t)$, which is then used to estimate the clean sample: $\hat{\mathbf{x}}_0(\mathbf{x}_t) = \frac{1}{\sqrt{\bar{\alpha}_t}}\left(\mathbf{x}_t - \sqrt{1 - \bar{\alpha}_t}\,\epsilon_\theta(\mathbf{x}_t, t)\right)$. Using this estimate, the previous latent state is computed via update:

$$\mathbf{x}_{t-1} = \sqrt{\bar{\alpha}_{t-1}}\,\hat{\mathbf{x}}_0(\mathbf{x}_t) + \sqrt{1 - \bar{\alpha}_{t-1} - \sigma_t^2}\cdot\epsilon_\theta(\mathbf{x}_t, t) + \sigma_t\mathbf{z}, \quad \mathbf{z} \sim \mathcal{N}(0, \mathbf{I}),$$

where $\sigma_t$ controls the level of stochasticity. The reverse process can be parameterized in various equivalent forms, including noise prediction $\epsilon_\theta(\mathbf{x}_t, t)$, posterior mean prediction $\mu_\theta(\mathbf{x}_t, t)$, or direct clean-sample prediction $\hat{\mathbf{x}}_0(\mathbf{x}_t)$.

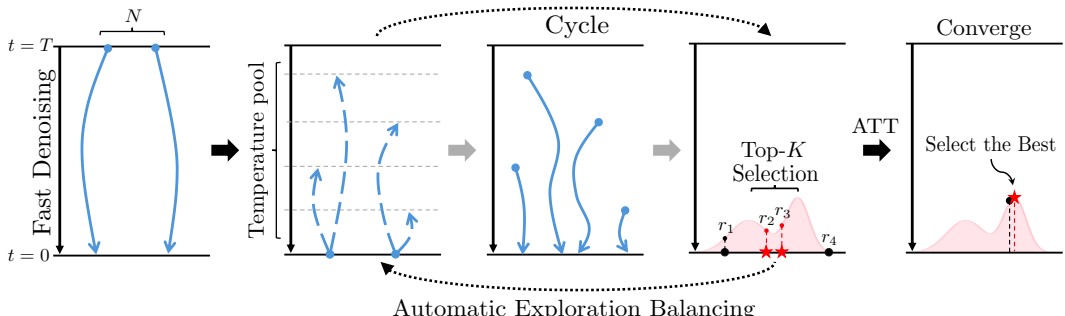

**Figure 1: Overview of our method.** (1) Start with $N$ particles, use jumpy denoising $x_T \rightarrow x_{T-j} \rightarrow \cdots \rightarrow x_0$. (2) Replicate each particles $J$ times and send each particles to multiple noise levels (AEEB). (3) Re-denoise particles. (4) Select Top $K$ particles. (5) We repeat steps (2)–(4) until the adaptive terminal condition is satisfied (CDS). The number of cycles executed in this way defines the Adaptive Thinking Time (ATT).

**Reward-guided generation.** In many practical scenarios, it is desirable not only to generate realistic samples from a diffusion model, but also to align them with a predefined reward function $r(\mathbf{x}) \in \mathbb{R}$. A common objective is to sample from a distribution that biases toward high-reward samples while staying close to the original model distribution [28]. This can be formalized as:

$$p^{(\alpha)}(\mathbf{x}) \propto \exp(r(\mathbf{x})/\alpha) \cdot p_{\text{pre}}(\mathbf{x}),$$

where $\alpha > 0$ is a temperature parameter controlling the trade-off between reward optimization and fidelity to the pre-trained distribution $p_{\text{pre}}$. As $\alpha \rightarrow 0$, the distribution becomes increasingly concentrated on reward-maximizing samples [15].

## 3 ABCD: Adaptive Bi-directional Cyclic Diffusion

We propose Adaptive Bi-directional Cyclic Diffusion (ABCD), a novel inference framework for diffusion models that enables *adaptive and efficient* scaling with respect to the available computation budget. ABCD is designed to flexibly trade off exploration and exploitation during inference and to dynamically determine when to terminate. The framework is composed of three core components: (i) **Cyclic Diffusion Search**, (ii) **Automatic Exploration**, and (iii) **Adaptive Thinking Time**.

### 3.1 Cyclic Diffusion Search

The core idea of Cyclic Diffusion Search is to enable iterative inference by cycling bi-directionally through the diffusion timeline, alternating between the denoising (reverse) and noising (forward) processes. This approach allows the model to refine its predictions over time while adaptively revisiting earlier stages of the generative process to explore alternative solution paths. Each iteration of the cycle consists of three main stages: Denoising, Selection-and-Copy, and Noising.

**Denoising.** As a particle-based method, the inference process begins by sampling a set of $N$ particles from the standard Gaussian prior distribution $\mathcal{N}(0, \mathbf{I})$. Leveraging the pretrained diffusion model, each particle is then denoised from $t = T$ to $t = 0$ using a coarse, accelerated Denoising Diffusion Implicit Models (DDIM) [26] trajectory that skips intermediate steps. This allows us to quickly obtain initial sample estimates $\mathbf{x}_0$ from the noisy latents $\mathbf{x}_T$, significantly reducing the time to acquire a preliminary solution candidate. While this fast DDIM process may yield suboptimal initial outputs, the key idea of our framework is to mitigate this through subsequent cyclic refinement. Unlike some prior approaches [16] that interleave intermediate noising steps during the denoising pass, we prioritize reaching $t = 0$ quickly to establish a diverse set of starting points for the cyclic search.

**Selection and copy.** Once all particles reach $t = 0$, we evaluate them using a task-specific reward function, yielding scores $r_1, r_2, \ldots, r_N$. To guide the subsequent search towards promising regions, we select the top-$K$ particles with the highest reward scores. These $K$ selected particles serve as "anchor" points for the next phase of exploration. Each of these $K$ particles is then replicated $J$ times, producing a total of $K \times J$ particles.

---

**Algorithm 1** ABCD: Adaptive Bi-directional Cyclic Diffusion

---

1: **procedure** ABCD(`verifier`, $T$, $\mathcal{T}$, `max_iter`, $\kappa$, $N$, $K$, $J$)
2:     Initialize $N$ particles $\{x_T^{(i)}\}$ from $\mathcal{N}(0, I)$
3:     Denoise to obtain $\{x_0^{(i)}\}$ via Fast denoising
4:     **while** less than `max_iter` **do**
5:         **Selection**: Select top-$K$ particles using `verifier`
6:         **If** Selected top-$K$ particles come from smallest $t_g$ for $\kappa$ times consecutively
7:             **break**
8:         **Copy**: Replicate each $K$ particles $J$ time
9:         **Noising**: Send each $K$ particles to each $t_g \in \mathcal{T}$
10:        **Denoising**: Denoise from each $t_g$ *go-back* temperature
11:    **end while**
12:    **return** Select best particle from top-$K$ particles
13: **end procedure**

---

**Noising.** The replicated particles are then sent back to an earlier diffusion step $0 \leq t' \leq T$—referred to as the *go-back* timestep—via the noising process of the diffusion model: $q(\mathbf{x}_{t'}|\mathbf{x}_0)$. This operation reintroduces stochasticity and enables the particle to re-enter the denoising trajectory from a different point in the diffusion space. The full cycle—denoising, selection-copy, and noising—is then repeated iteratively until a predefined stopping criterion is met.

## 3.2 Automatic Exploration-Exploitation Balancing

A key limitation of the basic Cyclic Diffusion Search, as described above, lies in its reliance on a fixed *go-back* timestep, treated as a manually specified hyperparameter. In principle, there may exist a *go-back* step that optimally balances exploration and exploitation at each stage of inference. However, identifying this optimal step is generally difficult and time-consuming. Moreover, our experiments show that the optimal *go-back* step not only varies across different instances but could also evolve dynamically throughout the inference trajectory of a single instance. Consequently, relying on a fixed *go-back* step is both hard to tune and fundamentally suboptimal for adaptive inference.

**Illustrative example.** Consider solving a Sudoku puzzle. In the early stages, when only a few digits are provided, global exploration is beneficial to fill in plausible candidates and explore a wide solution space. In contrast, in the later stages—when most of the board is correctly filled—aggressive global changes can corrupt nearly-complete solutions. At that point, cautious local refinement is preferable. This illustrates the need to dynamically adjust the exploration-exploitation balance during inference.

**Implementation.** To implement automatic exploration mechanism, we modify the noising step of the previously described cyclic diffusion search as follows. We begin by defining a set of predefined *go-back* timesteps, referred to as the "temperature pool", denoted by $\mathcal{T} = (t_1, t_2, \ldots, t_M)$, where each $t_i$ corresponds to a distinct level of exploration. A simple and effective way to construct $\mathcal{T}$ is to uniformly partition the diffusion timeline into $M$ segments. After the select-and-copy step, yielding $K \times J$ particles, the replicas are distributed across the predefined temperatures in $\mathcal{T}$ via the forward (noising) process, rather than being sent to a single *go-back* timestep. This allows each anchor particle to probe multiple exploration depths in parallel, enabling the system to implicitly identify effective *go-back* steps through iterative feedback. After completing the re-noising, all $K \times J$ particles are denoised back to $t = 0$, and a new top-$K$ set is selected. This way, particles from suboptimal temperatures will automatically be discarded. This cycle continues until a stopping criterion is satisfied. See Figure 1 for an illustration of the algorithm.

## 3.3 Adaptive Thinking Time

When should we stop this cyclic inference computation? Unlike single-pass inference methods that terminate once reaching $t = 0$, ABCD's cyclic structure raises a nontrivial question of when to conclude the iterative search. A naive solution is to impose a fixed computation budget (e.g., a maximum number of cycles), but this risks either underutilizing available compute or wasting resources when further refinement yields no benefit.

To address this, ABCD incorporates an implicit uncertainty measure derived from the temperature distribution of the top-$K$ particles. After each cycle, we record the re-noising temperatures from which the current top-$K$ particles originated. If high-temperature particles are present in the top-$K$, this suggests that exploration is still yielding valuable improvements. Conversely, if all top-$K$ particles originate from the lowest temperature, it implies that global exploration has little impact and the search is now confined to local refinement—signaling that the process may have converged.

As a practical implementation of this idea, we adopt a simple yet effective stopping rule: terminate the inference when all top-$K$ particles originate from the lowest temperature for $\kappa$ consecutive cycles. The hyperparameter $\kappa$ controls the sensitivity of the termination criterion, balancing early stopping against the opportunity for further refinement.

## 4   Related works

**Inference-time scaling with pre-trained diffusion models.** There is growing interest in leveraging pre-trained diffusion models more effectively during inference, by adaptively steering the generation process during sampling. A unified perspective [3, 28] frames this as sampling from a reward-augmented distribution of the form $p(\mathbf{x}) \exp(\lambda r(\mathbf{x}))$. A common approach is classifier guidance [5, 10], which assumes access to a differentiable value function. For non-differentiable rewards, derivative-free strategies such as SVDD [15], Beam Search [17], DSearch [14], and MCTS [35] have been proposed. Alternatively, sampling-based approaches like Importance Sampling and Sequential Monte Carlo (e.g., TDS [31], FKD [24]) offer principled reward-weighted trajectory selection. However, most methods operate under uni-directional denoising and rely on static search schedules.

**Planning and reasoning with diffusion models.** Diffusion models have recently gained traction for long-horizon planning and complex reasoning tasks. Diffuser [13] reinterprets planning as conditional generation over offline trajectories. Subsequent works [1, 4, 6, 36] extend this idea by using diffusion models as world models. On the reasoning side, recent studies explore diffusion models for structured inference. Energy-based diffusion models have been applied to symbolic reasoning tasks [7, 8, 35], and SRM [30] applies diffusion to visual spatial reasoning.

## 5   Experiments

We evaluate our proposed method on a diverse suite of challenging tasks that require generating samples in sparse or previously unexplored regions at training phase: (1) toy Mixture of Gaussian(MoG), (2) point-mass maze navigation [19], (3) Sudoku puzzle completion [18, 29], (4) path generation on unseen pixel maze image [12], (5) molecular structure prediction [11], and (6) text-to-image generation. These tasks present distinct challenges: MoG shows the importance of multiple *go-back* temperature pool as the proof-of-concept; point-mass maze requires long-horizon planning over 1000 steps; Sudoku demands logical consistency across row, column, and block constraints; path generation on unseen maze image tests generalization to novel environmental structures; molecular structure prediction requires generating valid 3D conformations under chemical and physical constraints; and image generation illustrates that our approach also operates effectively in the high-dimensional setting, demonstrating its scalability beyond structured low-dimensional tasks. Detailed task configurations are discussed in Appendix B.2.

### 5.1   Baselines

We compare ABCD against several strong representative baselines: **Base Diffusion**: The standard diffusion model without additional computational scaling, serving as our primary baseline. **Best-of-$N$ (BoN)**: A computationally scaled variant that generates $N$ independent samples and selects the best according to a task-specific verifier used consistently across all models. **Diffusion Beam Search (BS)** [17]: Combines diffusion with beam search to scale the denoising process, pruning unpromising candidates during generation. **Sequential Monte Carlo Diffusion (SMC)** [24, 31]: Employs resampling during the denoising process to focus computational resources on more promising trajectories. Implementation details were adapted from [24] to suit our experimental setting. **Search over Paths (SoP)** [16]: Enhances generation by allowing limited backward steps (adding noise) to promising samples, enabling exploration of wider solution spaces compared to strictly unidirectional approaches. Detailed Baseline configurations are discussed in Appendix B.3.

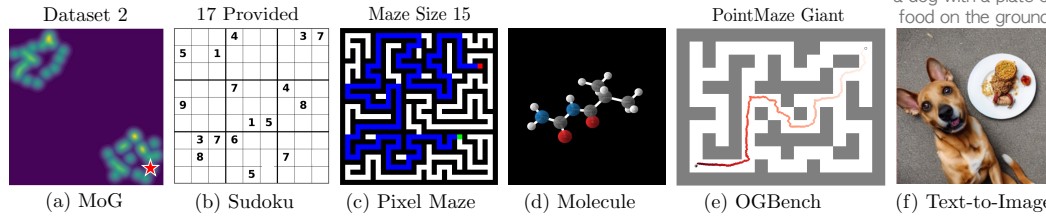

| Dataset 2 | 17 Provided | Maze Size 15 | | PointMaze Giant | a dog with a plate of food on the ground |
| (a) MoG | (b) Sudoku | (c) Pixel Maze | (d) Molecule | (e) OGBench | (f) Text-to-Image |

**Figure 2: Evaluated Tasks.** (a) Mixture of Gaussian for Proof-of-concept that we need multiple go back temperature. (b) Sudoku puzzle completion demanding logical consistency across multiple constraints simultaneously; (c) Pixel Maze Path Finding testing generalization to novel environmental structures not encountered during training; (d) Molecular structure prediction requiring physically and chemically valid 3D conformations. (e) OGBench PointMaze navigation requiring complex planning over 1000+ steps (f) Text-to-image generation requiring complex, high-dimensional outputs that are well aligned with the given textual descriptions;

**Table 1: Left: Success rate (%) and distance to the goal on MoG.** We compare multi *go-back* (ABCD) and fixed single *go-back* (GB 20, 40, 60, 80).

| Method | Dataset 1 | | Dataset 2 | |
| | Success | Distance | Success | Distance |
|---|---|---|---|---|
| ABCD (Ours) | **100** | **0.08 ± 0.0** | **100** | **0.03 ± 0.0** |
| GB 20 | 95 | 0.17 ± 0.0 | 26 | 3.97 ± 6.0 |
| GB 40 | 0 | 2.90 ± 0.6 | 59 | 1.73 ± 4.6 |
| GB 60 | 0 | 7.80 ± 0.8 | 7 | 2.94 ± 1.1 |
| GB 80 | 0 | 42.81 ± 6.6 | 0 | 36.46 ± 11.1 |

**Table 2: Right: Success rate (%) and mean iteration to reach the goal on MoG.** Every method terminates when it reaches the goal mode.

| Method | Dataset 1 | | Dataset 2 | |
| | Success | Mean Cyc | Success | Mean Cyc |
|---|---|---|---|---|
| ABCD (Ours) | **100** | 3.36 | **100** | 7.02 |
| GB 20 | **100** | **2.06** | 79 | 16.14 |
| GB 40 | 8 | 46.38 | **100** | **6.77** |
| GB 60 | 0 | – | **100** | 15.05 |
| GB 80 | 0 | – | 6 | 43.33 |

All baselines used the same pre-trained diffusion model per tasks, and inference-time budgets were expanded as much as possible while preserving their core operational characteristics. In BS and SoP, inference-time compute was increased by more frequent expansion and selection, while SMC scaled compute via higher resampling frequency, and BoN by increasing the number of particles $N$. For fair comparison, the total number of particles was set equal across all methods (except BoN). While we primarily report wall-clock time in the main paper to analyze inference-time scaling behavior, we additionally provide a analysis of the Number of Function Evaluations (NFEs) in Appendix B.16. Additional experimental details for each tasks are provided in the Appendix B.6, B.8, B.10, B.12.

## 5.2 Mixture of Gaussian

**Setup.** We construct two toy datasets to illustrate the need for multiple *go-back* noise levels and terminal condition during inference refinement (Figure 2(a)). Dataset 1 has modes clustered locally, so small-step refinement suffices; Dataset 2's modes lie in distant regions, requiring occasional large "jumps" to escape poor initial prediction. We train separate diffusion models on each and condition sampling on distance to a target (red star). Full details appear in Appendix B.4.

**Result.** We evaluate ABCD against four fixed single go-back temperatures (GB 20, 40, 60, 80) under two inference regimes: a fixed 25-cycle budget and an open-ended budget of up to 100 cycles with early stopping upon goal attainment. As shown in comparison with fixed 25 cycles (Table 1): only ABCD attains 100 % success and minimal final distance on both datasets. Fixed small *go-back* steps work on Dataset 1 but cannot escape poor initializations in Dataset 2, whereas fixed large steps succeed escaping from poor initializations on Dataset 2 but over-cycle on Dataset 1 and 2 (e.g., reaching the goal at cycle 20 but diverging by cycle 25 after losing useful information). As shown in comparison with up to 100 cycles with early stopping (Table 2): some fixed levels solve one dataset, but only ABCD consistently achieves both 100 % success with comparable average cycles. These findings underscore that exploration (coarse jumps) and exploitation (fine steps) require different *go-back* levels—a balance that ABCD's adaptive termination naturally realizes.

## 5.3 Sudoku puzzle completion

**Setup.** We evaluated ABCD against baseline methods on the Sudoku puzzle completion task [18, 29], a canonical logical-reasoning benchmark. In Sudoku, a 9×9 grid must be filled under row, column, and 3×3-block constraints, with each new grid filled based on the current grid state. An adaptive

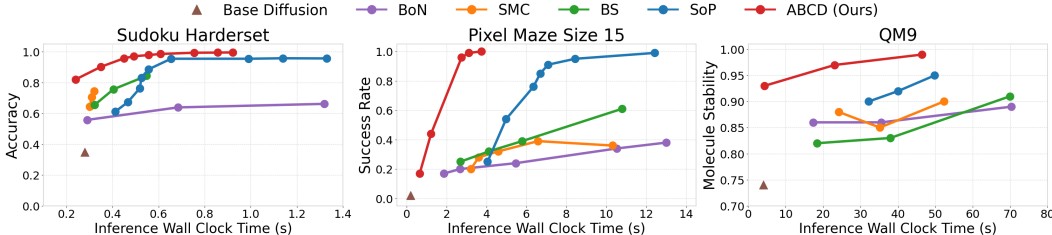

Figure 4: **Left**: **Sudoku Puzzle Completion result.** Mean accuracy on Harder dataset (17-28 entities provided) by giving more computational budget (sec.). **Middle**: **Pixel maze path finding result.** Success rate on the OOD Pixel Maze size-15 test set. **Right**: **Molecular 3D structure prediction task result.** Molecular Stability rate on the QM9 dataset. For each method, inference-time computation was scaled as much as possible along its core controllable axis for inference time scaling to ensure a fair comparison under expanded budgets.

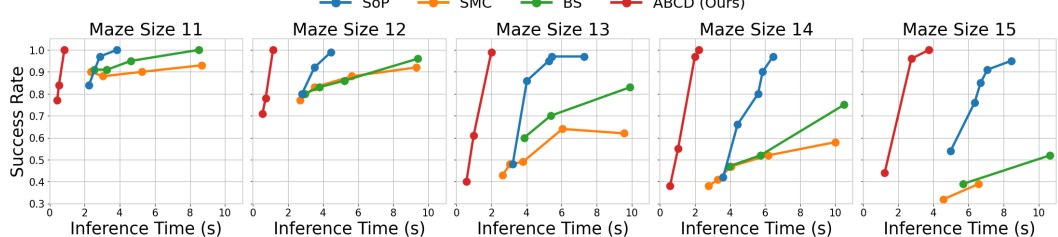

Figure 5: **Pixel Maze path finding results on multiple maze size.** Success rate comparison across maze sizes in the Pixel Maze task. All settings involve OOD evaluation, where models are trained on smaller mazes (sizes 4∼6) and tested on larger ones. As maze size increases, the search space grows and the performance gap between other methods widens, highlighting the importance of adaptive inference-time exploration in complex scenarios.

search strategy—broadly exploring when few clues are given and then locally refining once most entries are correct—proves particularly effective. We trained all models on puzzles with 31–42 given digits [29] and tested them on more challenging instances containing only 17–28 givens [18], thereby measuring each method's ability to produce logically consistent solutions. See Appendix B.8 for additional experimental details.

**Result.** Figure 4(Left) shows that ABCD consistently outperforms all baselines, achieving higher accuracy in less wall-clock time and displaying superior inference-time scaling. Notably, SoP [16] never achieves perfect accuracy (95.5 % accuracy), whereas ABCD attains 100 % accuracy in fewer cycles. Moreover, as illustrated in Figure 3, ABCD's accuracy remains stable across all difficulty levels, while baseline methods exhibit marked performance degradation on the hardest instances. We attribute this robustness to ABCD's dynamic computation allocation: it adaptively concentrates additional inference effort on more challenging puzzles, in contrast to baselines that expend a fixed, uniform budget.

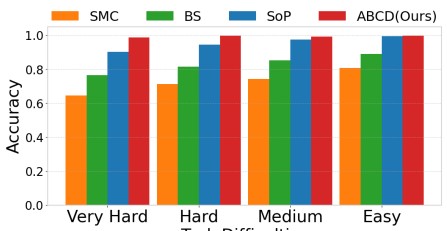

Figure 3: **Mean accuracy across four difficulty levels**—Very Hard (17-19 provided), Hard (20-21 provided), Medium (22-24 provided), and Easy (25-27 provided).

### 5.4 Path generation on unseen pixel maze image

**Setup.** We evaluated ABCD's adaptability on a path-generation task using mazes of previously unseen sizes [12]. Every method leveraged the same diffusion model pretrained on small mazes (sizes 4, 5, and 6) and was then tested on larger, structurally distinct mazes [35]. This setting rigorously probes each algorithm's capacity to generalize to novel spatial configurations at inference time. Additional experimental details are provided in Appendix B.10.

**Result.** As shown in Figure 4(Middle), ABCD reaches near-perfect success rates significantly faster than all baselines, evidencing a superior time–accuracy trade-off. In contrast, most competing

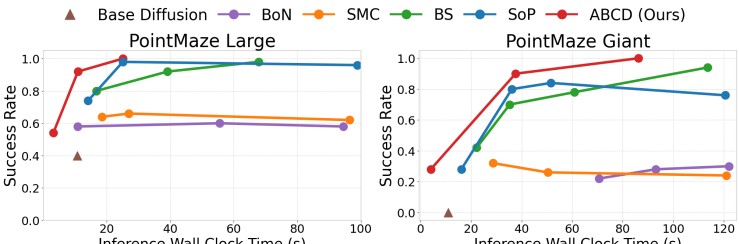

**Figure 6: OGBench PointMaze task results.** Success rates across large and giant mazes. Our ABCD approach consistently outperforms baselines, rapidly achieving higher success rates with fewer inference time and attaining perfect performance, particularly in the Giant maze, emphasizing its superior efficiency.

methods—including Base Diffusion, BoN, BS, and SMC—never exceed a 60 % success rate even under generous computational budgets. ABCD, by comparison, attains 100 % success in substantially less time. Figure 5 further examines performance across maze sizes 11–15 in the Pixel Maze task. Here again, ABCD consistently dominates both success rate and inference time at every size, and the margin over baselines widens as the search space grows. These findings underscore the value of adaptive, instance-specific exploration strategies especially in complex search demanding tasks.

## 5.5   Molecule generation

**Setup.** We evaluate ABCD on the 3D molecule generation task using the QM9 dataset [20], which contains 130k small molecules with annotated properties and atomic coordinates. For our evaluation, we utilized the Equivariant Diffusion Model (EDM) [11] as the pre-trained model. We then applied various inference-time methods to generate valid 3D conformations.

**Result.** As shown in the Figure 4(Right), ABCD significantly outperforms all baselines by achieving higher molecular stability with less inference wall-clock time. For instance, ABCD reaches a molecule stability of 0.97 within approximately 20 seconds, a threshold that other baselines either fail to attain or require considerably more computational time to approach. Notably, our method achieved a peak molecule stability of approximately 0.99, which is about 5.32% higher than the peak stability of 0.94 achieved by SoP [16], the next best-performing baseline. Furthermore, the performance gap between ABCD and the baselines widens at higher stability levels, emphasizing the robustness of our method in complex molecular generation tasks. See Appendix B.12 for more details.

## 5.6   OGBench Point Maze

**Setup.** We further evaluate the ABCD framework using OGBench's PointMaze environments [19]. In this task, agents must navigate to a specified goal region. The base diffusion model follows Diffuser [13], and is trained using the offline dataset provided by OGBench [19], containing trajectories collected by a noisy expert policy. Each environment includes 5 distinct tasks with different start-goal pairs. At inference time, the goal information is provided to the verifier for scoring the particles. To let the model efficiently plan over long horizon while reaching the goal as quickly as possible, we employed a value guidance function [5] to minimize the distance from the final state to the goal.

**Result.** As shown in Figure 6, ABCD consistently surpasses other baselines. Moreover, only ABCD achieves perfect performance on both mazes. It reflects the advantages of our method: by quickly denoising to $t = 0$, it locates valid trajectories almost immediately, rather than spending compute on uniformly long reverse sampling path. In maze giant, the gap widens because larger mazes demand more on both global exploration (to reach the goal) and fine refinement (to thread tight corridors). Further results analysis and visualization of the trajectories can be founded in the Appendix B.7.

## 5.7   Text-to-image generation

**Setup.** Finally, we evaluate ABCD on the image generation tasks. We adopt Stable Diffusion v1.5, a latent diffusion model, to serve as the image prior for generating $512 \times 512$ samples $x \sim p_\theta(x|y)$, conditioned on the textual prompt y. For downstream reward function, we utilize metrics including compressibility, aesthetic evaluation (using LAION's V2 Aesthetic Predictor [22]), and human

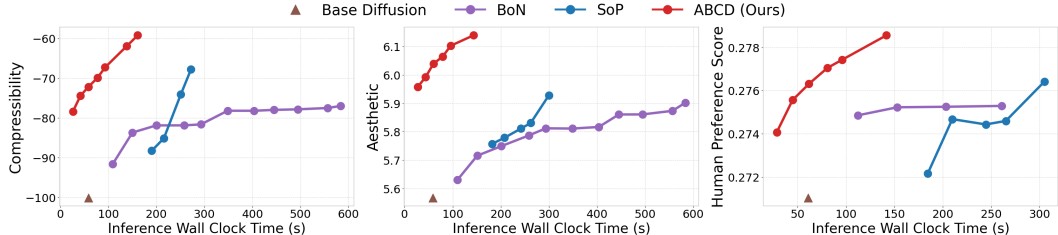

**Figure 7: Text-to-image generation result.** Average reward with respect to compressibility, aesthetic score and human preference score.

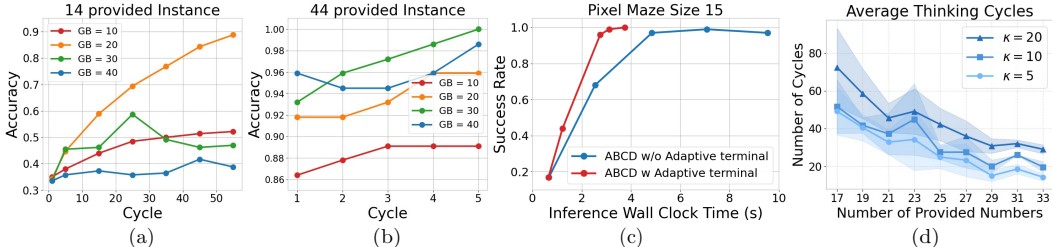

**Figure 8: (a-b) Per-instance analysis of inference-time scaling behavior across different go-back temperatures. (a)** Results for Sudoku instances with 14 provided entities. **(b)** Results for instances with 44 provided entities. **(c) Performance comparison with vs. without adaptive terminal condition** in Pixel Maze Size 15. x-axis is the average inference time per sample. **(d) Different thinking assignment** per hardness in Sudoku (provided number from 17 to 33).

preference evaluation (HPSv2 from [32]) for comprehensive evaluation. Further details regarding the experiments can be found in Appendix B.14.

**Result.** Figure 7 shows that ABCD consistently outperforms two strong baselines, BoN and SoP, in generating high-reward samples. It efficiently navigates the high-dimensional image space to identify high-reward samples. As computational budget increases, all methods see reward improvements, but ABCD benefits the most. For example, in all metrics, as we double or triple the inference time, BoN barely climbs, while SoP achieves moderate gains. By contrast, ABCD can hit the same compression level that SoP reaches at 272s in less than a quarter of the time. In addition, to reach a human preference score of 0.2764, SoP needs over 300s, while ABCD does it in under 81s. These highlights that its instance-specific exploration strategy effectively leverages additional computation to better align generated samples. Visualizations of the generated images are provided in Appendix B.15.1.

### 5.8 Do we need to care about *go-back* temperature level?

**Instance-level optimal temperature.** Figure 8(a–b) plots cycle-wise accuracy for two Sudoku instances (14 vs. 44 givens) under fixed go-back noise levels. Certain noise settings plateau after only a few cycles and the plateau point differs between puzzles, indicating that each instance possesses its own optimal noise level. (see Appendix C.1).

**Temperature selection per cycle.** Figure 9 tracks the provenance of selected particles source across cycles for three instance types in Sudoku (17, 19, and 32 givens). Harder puzzles continue sampling from a broad range of noise levels, while easier ones converge rapidly to low-variance perturbations. Early iterations emphasize high-variance (global) moves; later iterations favor low-variance (local) refinements. These patterns demonstrate ABCD's dynamic, per-instance generative process through adaptive exploration vs. exploitation.

### 5.9 Do we need adaptive terminal condition?

**Efficiency of adaptive terminal condition.** An ablation study on Pixel maze size 15 (Figure 8(c)) compares fixed cycle length inference (blue) to our adaptive terminal criterion (red) with the same temperature pool. The adaptive policy stops early on simple puzzles—avoiding wasted compute—yet allocates extra iterations to hard ones, yielding consistently superior time–success trade-offs.

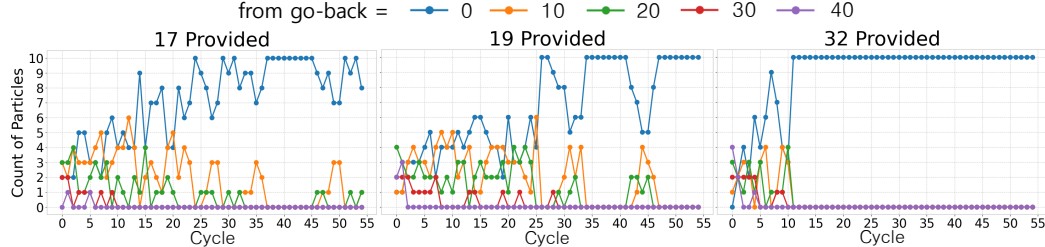

**Figure 9: The origin of the top-10 selected particles at each cycle.** The above plot shows, for different Sudoku difficulty cases, the origin of the Top-10 selected particles at each cycle. We observe that the behavior varies significantly across cases, and harder instances tend to require longer search. Additionally, early iterations tend to select samples with large modifications, while later cycles focus on smaller refinements. This indicates that our model automatically expands and adjusts the diffusion generation process, resulting in a different generation graph for each final $\mathbf{x}_0$ prediction.

**ABCD automatically assigns proper thinking time.** Figure 8(d) demonstrates that ABCD allocates more thinking cycles to harder puzzles (fewer givens) and terminates earlier on simpler ones, validating that our adaptive terminal criterion scales compute with problem difficulty. Furthermore, the stop-flag hyperparameter $\kappa$ directly controls exploration depth: larger $\kappa$ values enforce additional cycles before stopping, which boosts success on challenging instances. Identical trends hold on the Pixel Maze task. (see Appendix B.11, C.2,C.3, E for more detailed analysis).

## 6  Limitations and discussion

While ABCD performs well on moderately sized planning and reasoning tasks, its scalability to high-dimensional output spaces—such as high-resolution image or video generation—remains unexplored. Extending ABCD to such domains may require structural priors or hierarchical variants to maintain tractability and efficiency. In addition, although ABCD demonstrates strong empirical performance through its search-based inference strategy, its theoretical properties remain largely unexamined. A deeper theoretical understanding could be gained by connecting ABCD to principled frameworks such as Bayesian optimization or approximate inference.

A particularly promising future direction lies in amortizing the search process. The optimal re-noising steps and decisions discovered during ABCD's iterative inference could be treated as pseudo-labels for training a context-aware policy that predicts adaptive search strategies conditioned on the input. Such a learned policy could retain the adaptivity of ABCD while significantly reducing inference-time cost by eliminating online search. We leave the development of such amortized inference mechanisms to future work.

## 7  Conclusion

We introduced Adaptive Bi-directional Cyclic Diffusion (ABCD), a novel inference framework that enables instance-aware, compute-adaptive generation with diffusion models. By framing inference as a flexible search process, ABCD dynamically allocates computation through bi-directional diffusion cycles, adaptive exploration, and principled stopping. Across diverse domains—including planning, reasoning, and molecule generation—ABCD improves performance and efficiency over fixed-schedule baselines. This work underscores the promise of adaptive inference-time scaling.

## Acknowledgments and Disclosure of Funding

We thank to Junyeob Baek, Doojin Baek, and Hyeonseo Cho for insightful discussions and assistance with this project. This research was supported by GRDC (Global Research Development Center) Cooperative Hub Program (RS-2024-00436165) and Brain Pool Plus Program (No. 2021H1D3A2A03103645) through the National Research Foundation of Korea (NRF) funded by the Ministry of Science and ICT. The authors of Mila acknowledge the support from CIFAR, NSERC, the Future of Life Institute.

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

# A    Ethic statement

Our method is a general-purpose inference-time optimization framework designed to maximize task-specific reward functions. The ethical implications of its use are inherently tied to how the reward function is defined. While this flexibility allows for beneficial applications—enabling faster and more targeted generation aligned with desirable outcomes—it also raises the potential for misuse if the reward function encodes harmful or biased objectives. Therefore, the responsibility for ethical deployment lies in the careful design and oversight of the reward specification.

# B    Experiment details

## B.1    Computational resource

Our implementation is built using the PyTorch framework. The experiments are conducted on 2 machines: Ubuntu 20.04 machine equipped with an Intel(R) Xeon(R) Gold 6348 CPU @ 2.60GHz with 112 cores, 384 GB RAM, and NVIDIA GeForce RTX 4090 GPUs; Ubuntu 22.04 machine equipped with an Intel(R) Xeon(R) Gold 6230R CPU @ 2.10GHz with 104 cores, 256 GB RAM, and NVIDIA GeForce RTX 4090 GPUs. Each experiment is run individually on an NVIDIA GeForce RTX 4090 GPU.

## B.2    Task detail

We introduce the setting of experiments and diffusion models we used. For **Mixture of Gaussian**, we constructed the two datasets generated from the mixture of gaussians. Dataset 1 is generated from 1 region of Gaussians and Dataset 2 is generated from 2 region of the islands as you can see on the Figure 2(a). After pre-training two diffusion models on each dataset, we guide sampling at inference time by providing the distance to a target goal as a reward, encouraging generation toward the desired goal state. For **OGBench Maze**, we adopted the task setups from [19], running 5 sparse reward navigation tasks in each environment, each can take hundreds of steps to accomplish. The model is trained on standard dataset collected by noisy expert policy that repeatedly reaching randomly sampled goals. At test time, the model is required to generate proper plan using guidance from value function. For **Sudoku**, the basic setting is adopted from IRED [8], where the diffusion model is trained on SAT-Net dataset with 31 to 42 provided samples [29] and tested on harder RRN dataset [18] with 17 to 28 provided samples. For **Pixel Maze**, the basic setting is adopted from T-SCEND [35], where the dataset are generated by `maze-dataset` [12], and the diffusion model is trained with Maze sizes of $4 \times 4$ to $6 \times 6$ and tested with size of $11 \times 11$ to $15 \times 15$. For **Molecule Generation**, the basic setting is adopted from EDM [11]. The diffusion model is trained to generate 3D molecular geometries using the QM9 dataset [20], which contains approximately 130k small molecules, each with up to 9 heavy atoms. We utilized the standard splits of this dataset for training, validation, and testing. For **Text-to-Image generation**, we use Stable Diffusion v1.5—a widely adopted text-to-image diffusion model—as our pretrained backbone.

**Verifiers.** For **OGBench Maze**, we employ a soft verifier that scores each generated plan based on the proportion of collision-free states throughout the trajectory and the proximity to the goal at the end of the trajectory. For **Sudoku**, we use the ratio of correct answer that matches the ground truth label. For **Pixel Maze**, we use weighted sum of Precision, Recall, F1 score with rate of success score which indicates whether the path reaches the goal without passing the wall. For **Molecule Generation**, we use the Negative Log-Likelihood (NLL) to evaluate the generated molecular structures. For **Text-to-Image generation**, we use three verifier scores: compressibility [2], aesthetic [22], and human preference score [33].

**Evaluation metrics**. For **OGBench Maze**, we used the success rate (whether the planned trajectory reaches to the goal or not) of the actual rollouts from the generated trajectory plans. For **Sudoku**, we use the ratio of correct answer that matches the ground truth label. For **Pixel Maze**, we use success score (0 or 1) which indicates whether the path reaches the goal without passing the wall. For **Molecule Generation**, we use molecule stability (0 or 1), the proportion of generated molecules for which all atoms are stable. For **Text-to-Image generation**, we use three evaluation metrics: compressibility  [2], aesthetic [22], and human preference score [33]—which correspond to specific verifiers, respectively.

## B.3 Baseline detail

We compare ABCD (ours) to several representative methods capable of inference-time scaling.

**Base Diffusion** is the most basic baseline, which generates samples without any additional computation during inference. When a differentiable value function is available, we include **Classifier Guidance** [5] as a base diffusion baseline.

**Best-of-N(BoN)** generates $N$ samples from the pre-trained diffusion model and returns the sample with the highest reward score according to the verifier.

**Search over Paths (SoP)** [16] is a recently proposed method that performs inference-time scaling by alternating backward(noising) and forward(denoising) transitions in the denoising space. Starting from $M$ particles, the method perturbs each particle with $K$ different noise moving to $\Delta f$ steps backward, resulting in $M \times K$ expanded candidates (expansion). Each of these is then denoised forward for $\Delta b$ steps. After reaching the new states, each particles are scored using a verifier, and the top $M$ are selected to continue (selection). This process is repeated until the particles reach $t = 0$, with the constraint that $\Delta f < \Delta b$ to ensure overall forward progression.

**Diffusion Beam Search (BS)** [17] is another baseline that performs inference-time search over denoising trajectories. Starting from $M$ initial particles, the method branches each into $K$ candidate continuations every $p$ denoising steps, and retains the top $M$ candidates based on reward estimates. This beam-style selection continues until the denoising process reaches $x_0$. Our implementation is inspired by [17], with necessary modifications for our setting.

**Sequential Monte Carlo Diffusion (SMC)**[24] is a sampling-based method that employs Sequential Monte Carlo (SMC) to steer diffusion trajectories toward high-reward regions[31]. Inspired by the Feynman-Kac Steering Diffusion framework (FKD) [24], it begins with an initial set of $N$ proposal particles and performs iterative resampling every $p$ steps during the denoising process, using importance weights defined by tailored potentials. These potentials are designed to approximate the reward-weighted posterior distribution $p(\mathbf{x})p(c \mid \mathbf{x})$ (where the reward encodes the target condition), effectively guiding particle evolution toward desirable outputs. Our implementation follows the general structure of FKD, with modifications to accommodate our conditional generation setting. Specifically, we define $p(c \mid \mathbf{x})$ using the task-specific verifier score to guide the sampling process. As this is a sampling-based approach, we select the highest-scoring particle from the final set of $N$ samples drawn from the guided distribution.

**ABCD**'s inference-time termination is controlled by three hyperparameters: the percentage threshold, the maximum iteration bound (`max_iter`), and the persistence parameter $\kappa$. The percentage threshold defines the minimum fraction of top-$K$ particles that must originate from the zero *go-back* noise level in order to raise a termination flag. Inference is terminated once this condition holds for $\kappa$ consecutive iterations. We scale inference-time computation by varying both the percentage threshold and $\kappa$, which jointly determine the strictness of the stopping condition. Additionally, we adjust `max_iter` to cap the total number of iterations. As these hyperparameters increase, the termination criterion becomes more conservative, typically leading to longer inference time and improved performance through extended refinement.

Unless otherwise specified, in both SoP, BS and SMC, since we have reward function defined on $\mathbf{x}_0$ space, the reward evaluations at each step $\mathbf{x}_t$ are performed using a predicted clean sample $\hat{\mathbf{x}}_0(\mathbf{x}_t)$ (approximation of $\mathbb{E}_{\mathbf{x}_0 \sim p_{\text{pre}}}[\mathbf{x}_0 | \mathbf{x}_t]$), which is then passed into the verifier[14, 15, 24]. With this choice, we get the intermediate reward $r(\mathbf{x}_t, c) = r(\mathbf{x}_0 = \hat{\mathbf{x}}_t, c)$. This ensures that reward signals are consistent across trajectories and noise levels.

## B.4 MoG experiment detail

We constructed two Mixture of Gaussians datasets: Dataset 1 has 36 modes concentrated in a single region, while Dataset 2 distributes 36 modes across two regions (18 modes each). Two separate diffusion models were pretrained on each datasets using 100 denoising steps. In each experiment, we initialize with $M = 2$ particles sampled from the prior, and use $K = 4$ distinct noise samples to guide the particles backward.

## B.5 MoG result detail

### B.5.1 Small *go-back* levels vs. large *go-back* levels

As shown in Figure 10, small *go-back* levels effectively guide the particle toward the goal when navigating within a single region landscape. However, in the two-regions scenario (Figure 11), small *go-back* levels struggle to move particles across regions to reach the goal. By contrast, large *go-back* levels significantly improve cross-region transitions.

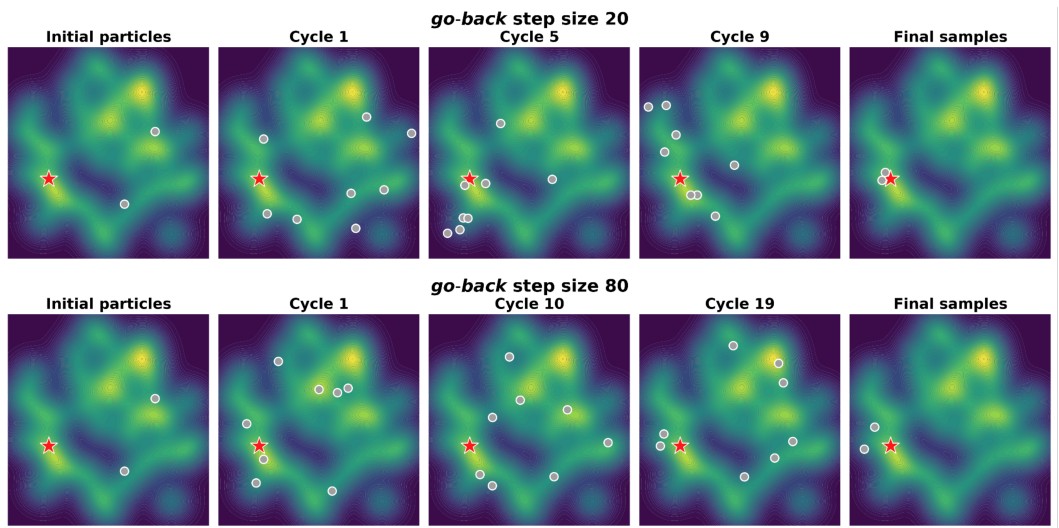

**Figure 10:** Compare different *go-back* step size on task 1.

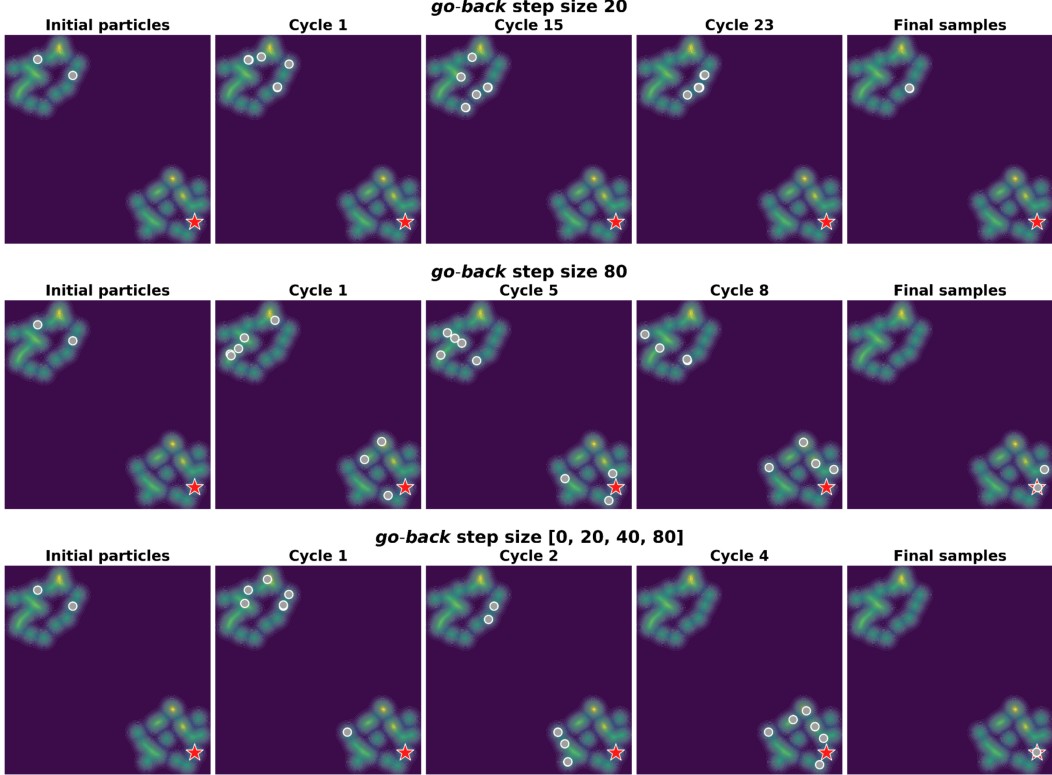

**Figure 11:** Compare single and multiple *go-back* step size on task 2.

### B.5.2 Single *go-back* levels vs. multiple *go-back* levels

After leveraging the large *go-back* to traverse between regions, switching to small *go-back* becomes advantageous for shifting from exploration to exploitation. This transition enables fine-grained goal-oriented refinement, as demonstrated in Figure 11. All instances begin from an initial incorrect prediction, requiring additional correction during inference. *go-back* $= 20$ fails to escape the initial cluster of modes, whereas *go-back* $= 80$ enables transitions across regions but refines inefficiently. Our method combines coarse transitions to explore distant regions and fine adjustments for local refinement, enabling faster and more effective inference time adjustment.

### B.6 OGBench Point Maze experiment detail

All methods were evaluated under the same setup, using 32 particles to ensure a fair comparison. For **SoP**, we used $M = 4$ and $K = 8$ in Maze Giant, and $M = 1$ and $K = 32$ in Maze Large. **BS** used $M = 8$ and $K = 4$ in both mazes, with a lookahead estimator [17] to have a better predicted $\hat{\mathbf{x}}_0(\mathbf{x}_t)$ with value guidance. **SMC** [24] was implemented with POTENTIAL TYPE = "sum", $\lambda = 0.1$ and used $N = 32$ particles in both mazes. **ABCD** was configured with $N = 32$, $K = 2$ and $J = 16$ in Maze Giant and $N = 32$, $K = 1$ and $J = 32$ in Maze Large. The adaptive terminal condition was set once more than 90% of top-$K$ particles consistently originated from the zero noise level over $\kappa$ consecutive steps. We used $\kappa = 30$ in Maze Giant and $\kappa = 5$ in Maze Large. All methods used a base diffusion model trained with 256 denoising steps. ABCD employed jumpy denoising with a jump length of 10.

To scale inference compute, we increased the branching-and-selection frequency $p$ for BS and SMC. For BoN, we scaled by increasing $N$. For SoP, we varied the step size for back-and-forth moves, as smaller steps incur higher compute due to finer-grained trajectory updates. For ABCD, we scaled by increasing the number of `max_iter`. Inference time was measured as the average time required to generate and execute a single plan in the environment.

## B.7 OGBench Point Maze result

Figure 6 compares the performance using an adaptive guidance scheme that increases guidance as the diffusion step $t$ approaches 0. For reference, the performance of all methods under the standard guidance scheme is reported in Table 3 and Table 4. ABCD consistently outperforms all baselines, and is the only method to achieve a perfect success rate within the given compute budget.

Table 3: Comparison of inference-time strategies in OGBench Point Maze Giant.

| Inference Method | Add. Compute | Performance | Inference wall clock time (sec.) |
|---|---|---|---|
| Base Diffusion | No add | 12±13 | 41.02 |
| BoN | N = 32 | 36±15 | 43.12 |
| | N = 64 | 54±22 | 45.34 |
| | N = 96 | 48±18 | 45.38 |
| | N = 128 | 60±24 | 48.64 |
| | N = 192 | 64±15 | 53.78 |
| | N = 288 | 64±17 | 87.94 |
| | N = 544 | 76±20 | 169.98 |
| SoP | $\Delta f$=200, $\Delta b$=100 | 40±9 | 59.12 |
| | $\Delta f$=100, $\Delta b$=50 | 60±15 | 76.56 |
| | $\Delta f$=50, $\Delta b$=25 | 86±13 | 88.06 |
| | $\Delta f$=40, $\Delta b$=20 | 92±10 | 99.08 |
| | $\Delta f$=30, $\Delta b$=15 | 94±9 | 108.84 |
| | $\Delta f$=20, $\Delta b$=10 | 90±13 | 131.90 |
| | $\Delta f$=10, $\Delta b$=5 | 94±13 | 181.94 |
| | $\Delta f$=6, $\Delta b$=3 | 96±12 | 250.06 |
| | $\Delta f$=4, $\Delta b$=2 | 86±9 | 342.94 |
| | $\Delta f$=2, $\Delta b$=1 | 80±0 | 588.98 |
| SMC | $p = 50$ | 28±20 | 52.52 |
| | $p = 25$ | 20±13 | 62.48 |
| | $p = 20$ | 20±13 | 67.20 |
| | $p = 15$ | 22±14 | 73.82 |
| | $p = 10$ | 22±11 | 90.82 |
| | $p = 5$ | 18±19 | 142.48 |
| BS | $p = 50$ | 38±11 | 65.40 |
| | $p = 20$ | 30±10 | 92.30 |
| | $p = 15$ | 52±18 | 74.84 |
| | $p = 10$ | 46±16 | 122.08 |
| | $p = 5$ | 50±16 | 239.12 |
| | $p = 1$ | 72±16 | 1014.30 |
| ABCD | max_iter = 1 | 82±11 | 12.70 |
| | max_iter = 5 | 94±9 | 34.02 |
| | max_iter = 10 | 96±8 | 65.52 |
| | max_iter = 15 | 96±8 | 85.98 |
| | max_iter = 20 | 96±8 | 117.54 |
| | max_iter = 25 | 96±8 | 138.04 |
| | max_iter = 30 | 98±6 | 184.40 |
| | max_iter = 35 | 100±0 | 202.58 |
| | max_iter = 40 | 100±0 | 214.08 |
| | max_iter = 45 | 100±0 | 271.68 |
| | max_iter = 50 | 100±0 | 295.12 |

**Table 4:** Comparison of inference-time strategies in OGBench Point Maze Large.

| Inference Method | Add. Compute | Performance | Inference wall clock time (sec.) |
|---|---|---|---|
| Base Diffusion | No add | 38±11 | 25.10 |
| BoN | N = 2 | 62±11 | 28.88 |
| | N = 4 | 70±16 | 34.82 |
| | N = 8 | 80±9 | 41.58 |
| | N = 16 | 88±10 | 39.66 |
| | N = 32 | 94±9 | 41.74 |
| | N = 64 | 98±6 | 40.12 |
| SoP | $\Delta f$=200, $\Delta b$=100 | 64±15 | 58.22 |
| | $\Delta f$=100, $\Delta b$=50 | 76±12 | 78.36 |
| | $\Delta f$=50, $\Delta b$=25 | 86±9 | 92.52 |
| | $\Delta f$=40, $\Delta b$=20 | 80±15 | 94.28 |
| | $\Delta f$=30, $\Delta b$=15 | 92±10 | 94.32 |
| | $\Delta f$=20, $\Delta b$=10 | 98±6 | 121.12 |
| | $\Delta f$=10, $\Delta b$=5 | 96±8 | 158.48 |
| | $\Delta f$=6, $\Delta b$=3 | 96±8 | 214.18 |
| | $\Delta f$=4, $\Delta b$=2 | 98±6 | 285.60 |
| | $\Delta f$=2, $\Delta b$=1 | 94±9 | 323.36 |
| SMC | $p = 50$ | 84±8 | 44.82 |
| | $p = 25$ | 84±8 | 51.26 |
| | $p = 20$ | 82±11 | 50.92 |
| | $p = 15$ | 82±11 | 53.08 |
| | $p = 10$ | 76±15 | 61.40 |
| | $p = 5$ | 74±9 | 81.90 |
| BS | $p = 50$ | 88±10 | 58.44 |
| | $p = 20$ | 88±10 | 60.50 |
| | $p = 15$ | 84±12 | 89.16 |
| | $p = 10$ | 90±13 | 115.64 |
| | $p = 5$ | 88±10 | 158.30 |
| | $p = 1$ | 94±9 | 801.42 |
| ABCD | J = 32 | 100±0 | 10.52 |

### B.7.1 OGBench Point Maze visualization

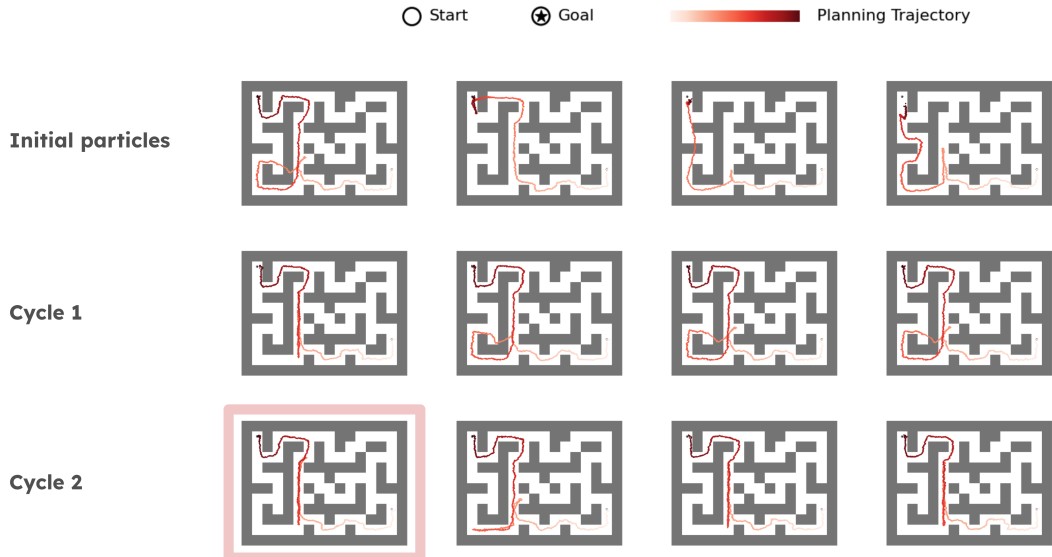

**Figure 12:** Visualization of the generated trajectories through cycle with ABCD. The figure highlighted with a red border indicates the selected plan for executing in the environment.

As shown in Figure 12, the initial trajectories proposed by jumpy denoising effectively capture a rough path from the start to the goal, yet contain noticeable invalid segments. Across subsequent cycles, Cyclic Diffusion Search progressively refines these trajectories by focusing on problematic regions and improves them toward viable plans. The highlighted plan selected for execution (red border) showcases ABCD's capability to preserve the nearly-complete solution by leveraging Automatic Exploration-Exploitation Balancing mechanism.

### B.8 Sudoku experiment detail

Experiments were conducted on 335 Sudoku test instances, each containing between 17 and 28 provided digits. Reported accuracy reflects the average performance across the full test set. Inference time was measured as the average time required to generate a single sample, computed by dividing the total time taken to generate all 335 outputs by 335. The base diffusion model was trained using $T = 50$ denoising steps. We follow the training configuration described in [35]. Each digit from 1 to 9 is represented as a one-hot vector, normalized, and passed through the diffusion model. During inference, the model outputs continuous predictions, which are then discretized by selecting the index with the highest predicted value as the final label.

To evaluate inference-time scaling strategies, each baseline was scaled along its primary computational axis while preserving its core algorithmic structure. **Base Diffusion** employed a single particle ($N = 1$). **BoN** increased inference cost by evaluating models with $N = 40, 480$, and $960$ particles. **BS** and **SoP** were implemented with $M = 4$ and $K = 10$. In **BS**, inference-time compute was scaled by increasing the branching-and-selection frequency $p$. **SoP** additionally scaled computation by reducing the step sizes for forward and backward transitions, $(\Delta b, \Delta f)$. **SMC** used $N = 40$ particles and scaled inference-time compute by adjusting the resampling frequency $p$ across timesteps. Following the FKD framework [24], we set the POTENTIAL TYPE to "max" and $\lambda = 100$ to enforce an optimization-style behavior. **ABCD (ours)** was configured with $N = 10$, $K = 10$, and $J = 5$, corresponding to four non-zero *go-back* noise levels (10, 20, 30, 40) and one zero level. The adaptive terminal condition terminated inference when a specified percentage of top-$N$ particles consistently originated from the zero noise level for $\kappa$ consecutive steps. We fixed `max_iter` to 100 and scaled ABCD's inference time by varying both the persistence parameter $\kappa \in \{1, 2, 3, 5, 10, 15, 20\}$ and percentage thresholds $\in \{0.6, 0.8, 1.0\}$. For fast denoising, ABCD employed five total inference steps, each performing a 10-step jump in the diffusion space. Please refer to Table 5 for detailed results.

## B.9 Sudoku result

We provide detailed results in Table 5 corresponding to the main Figure 4(Middle) presented in the main paper in this section. Additionally, Section B.9.1 visualizes the evolution of $\mathbf{x}_0$ predictions across refinement cycles in the Sudoku task, illustrating how different *go-back* temperature step sizes affect the iterative denoising trajectory.

**Table 5:** Comparison of various inference-time strategies across compute and performance in the Sudoku harder test set setting (from 17 to 28 provided entities). For each method, inference-time compute was scaled by adjusting its primary control axis—such as branching frequency, resampling rate, or particle count—while preserving its core algorithmic behavior.

| Inference Method | Add. Compute | Accuracy | Clock Time (sec.) | NFE |
|---|---|---|---|---|
| Base Diffusion | No add | 0.35 ± 0.1 | 0.28 | 50 |
| BoN | $N = 40$ | 0.55 ± 0.49 | 0.29 | 2000 |
| | $N = 480$ | 0.63 ± 0.48 | 0.68 | 24000 |
| | $N = 960$ | 0.66 ± 0.09 | 1.32 | 48000 |
| SMC | $p = 5$ | 0.64 ± 0.1 | 0.3 | 2000 |
| | $p = 2$ | 0.70 ± 0.1 | 0.31 | 2000 |
| | $p = 1$ | 0.74 ± 0.1 | 0.32 | 2000 |
| BS | $p = 5$ | 0.65 ± 0.46 | 0.32 | 900 |
| | $p = 2$ | 0.75 ± 0.48 | 0.40 | 1500 |
| | $p = 1$ | 0.84 ± 0.08 | 0.54 | 2460 |
| SoP | $\Delta f = 40, \Delta b = 20$ | 0.61 ± 0.09 | 0.41 | 2940 |
| | $\Delta f = 20, \Delta b = 10$ | 0.67 ± 0.1 | 0.46 | 3420 |
| | $\Delta f = 10, \Delta b = 5$ | 0.76 ± 0.1 | 0.51 | 3860 |
| | $\Delta f = 6, \Delta b = 3$ | 0.83 ± 0.11 | 0.52 | 3940 |
| | $\Delta f = 4, \Delta b = 2$ | 0.88 ± 0.09 | 0.55 | 4220 |
| | $\Delta f = 2, \Delta b = 1$ | 0.95 ± 0.06 | 0.65 | 4980 |
| | $\Delta f = 4, \Delta b = 3$ | 0.955 ± 0.06 | 0.99 | 8260 |
| | $\Delta f = 5, \Delta b = 4$ | 0.958 ± 0.06 | 1.14 | 9900 |
| | $\Delta f = 6, \Delta b = 5$ | 0.957 ± 0.06 | 1.33 | 11540 |
| ABCD | percentage = 0.6, $\kappa = 1$ | 0.82 ± 0.14 | 0.24 | 1059 |
| | percentage = 0.8, $\kappa = 1$ | 0.90 ± 0.1 | 0.35 | 1667 |
| | percentage = 1.0, $\kappa = 1$ | 0.95 ± 0.06 | 0.45 | 2296 |
| | percentage = 1.0, $\kappa = 2$ | 0.97 ± 0.05 | 0.49 | 2607 |
| | percentage = 1.0, $\kappa = 3$ | 0.979 ± 0.04 | 0.55 | 2827 |
| | percentage = 1.0, $\kappa = 5$ | 0.986 ± 0.03 | 0.60 | 3205 |
| | percentage = 1.0, $\kappa = 10$ | 0.99 ± 0.01 | 0.75 | 3942 |
| | percentage = 1.0, $\kappa = 15$ | 0.995 ± 0.01 | 0.85 | 4499 |
| | percentage = 1.0, $\kappa = 20$ | 0.996 ± 0.01 | 0.92 | 5013 |

## B.9.1 Sudoku visualization showing that proper *go-back* is necessary

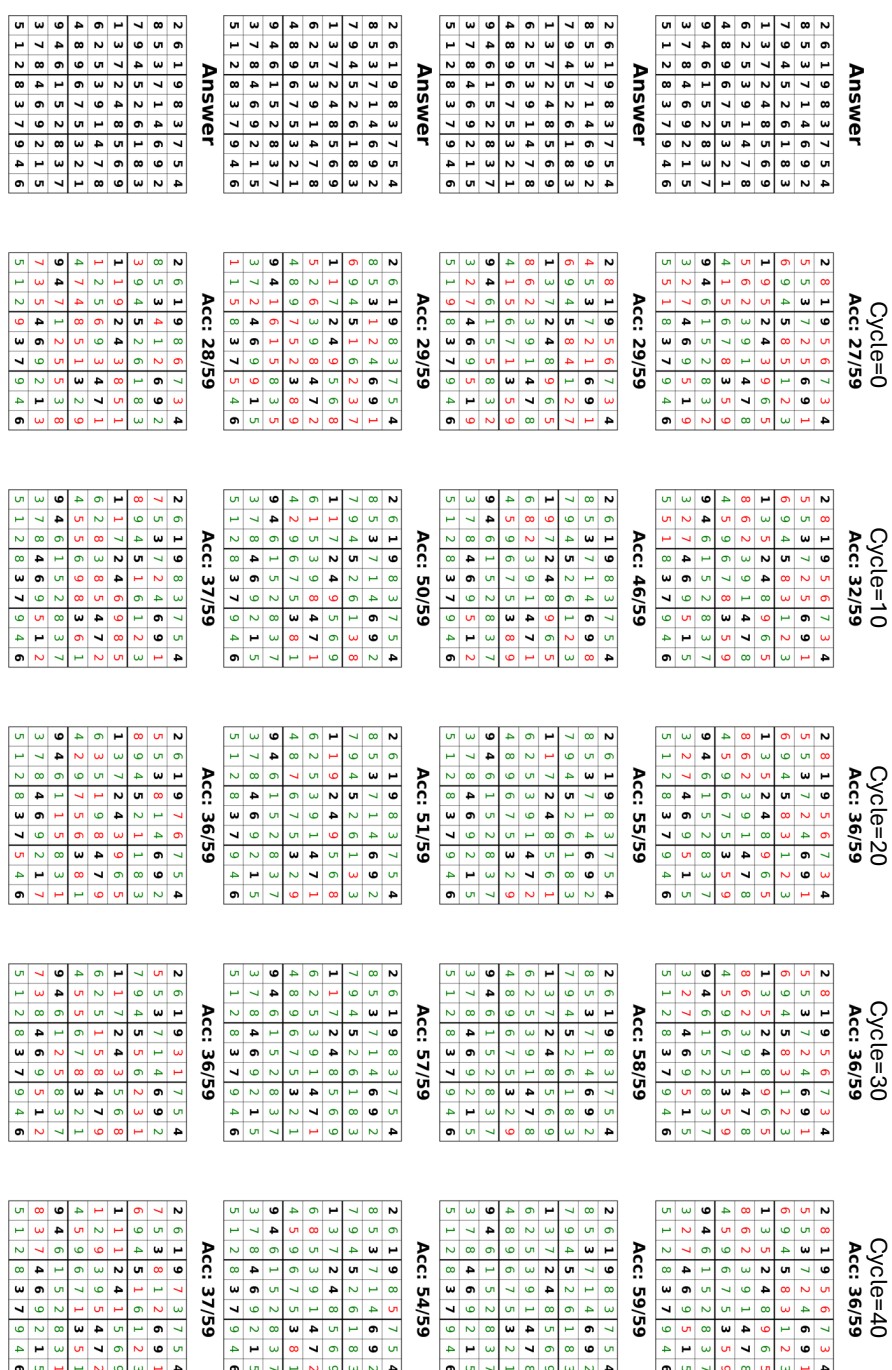

**Figure 13:** Visualization of $x_0$ predictions across cycles for four *go-back* step sizes. The rightmost column corresponds to *go-back* 10, followed by 20, 30, and 40 in ascending order from right to left.

As shown in the visualization results for Sudoku in Figure 13, when the *go-back* noise level is set to 10, the $x_0$ prediction remains unchanged despite repeated iterations failing to fix incorrect predictions. In contrast, with a noise level of 40, the $x_0$ prediction appears to update randomly at each iteration altering even the correct ones, as if generating entirely new guesses without accumulating information from previous predictions. For noise levels 20 and 30—especially 20—we observed meaningful performance improvement across iterations, where incorrect predictions were corrected while correct ones were unchanged. These findings confirm that appropriately selecting the *go-back* noise level—both per task and per instance—is crucial for maximizing the benefits of inference-time iterative refinement.

## B.10 Pixel Maze experiment details

Experiments were conducted on 100 Pixel Maze test instances for each maze size (from 11 to 15). The reported success rate reflects the proportion of cases in which the generated trajectory successfully reaches the goal without crossing any walls. Inference time was measured as the average time required to generate a single sample, computed by dividing the total generation time for 100 samples by 100. The base diffusion model was pretrained using $T = 50$ denoising steps. We follow the training configuration described in [35]. Both the maze layout (with walls encoded as 1 and empty spaces as 0) and the start/goal positions are represented as one-hot vectors and normalized before being passed into the diffusion model. During inference, the model produces continuous predictions, which are discretized by selecting the index with the highest value at each position, yielding the final binary path representation.

To assess inference-time scaling strategies, each baseline was scaled along its primary computational axis while preserving its core algorithmic behavior. **Base Diffusion** employed a single particle ($N = 1$). **BoN** increased inference cost by using a larger number of particles $N$. **BS** and **SoP** were implemented with $M = 4$ and $K = 10$. In **BS**, inference-time compute was scaled by increasing the branching and selection frequency $p$. In **SoP**, additional scaling was achieved by reducing the step sizes for forward and backward transitions, $(\Delta b, \Delta f)$. **SMC** used $N = 40$ particles and scaled inference-time compute by varying the resampling frequency $p$ over steps. Following FKD [24], we used POTENTIAL TYPE = "max" and set $\lambda = 100$ to emphasize optimization behavior. **ABCD (ours)** was configured with $N = 10$, $K = 10$, and $J = 5$, corresponding to four non-zero *go-back* noise levels (10, 20, 30, 40) and a zero level. The adaptive terminal condition terminated inference once a specified proportion of top-$N$ particles consistently originated from the zero noise level for $\kappa$ consecutive iterations. We scaled ABCD's inference time by varying $\kappa \in \{1, 2, 3\}$ and percentage thresholds $\in \{0.1, 0.2, 0.3, 0.6, 1.0\}$. To ensure computational efficiency, ABCD employed ten denoising steps, each executing a 5-step jump in the diffusion process. Please refer to Table 6 for detailed results.

## B.11 Pixel Maze result

We provide detailed results corresponding to the main Figure 4(Left) presented in the main paper in Section B.11.1, and further demonstrate that the varying inference-time behavior—previously observed in the Sudoku setting—also holds in the Pixel Maze setting, as discussed in Section B.11.2.

### B.11.1 Result table for Pixel Maze size = 15

This section provides a breakdown of the scaling factors we used, the performances and corresponding inference times for each method in Table 6.

**Table 6:** Comparison of various inference-time strategies across compute and performance in the Pixel Maze setting (size = 15). For each method, inference-time compute was scaled by adjusting its primary control axis—such as branching frequency, resampling rate, or particle count—while preserving its core algorithmic behavior.

| Inference Method | Add. Compute | Success rate | clock Time (sec.) | NFE |
|---|---|---|---|---|
| Base Diffusion | No add | 0.02 ± 0.13 | 0.2 | 50 |
| BoN | $N = 20$ | 0.17 ± 0.37 | 1.87 | 1000 |
| | $N = 40$ | 0.2 ± 0.4 | 2.67 | 2000 |
| | $N = 80$ | 0.24 ± 0.42 | 5.46 | 4000 |
| | $N = 160$ | 0.34 ± 0.47 | 10.52 | 8000 |
| | $N = 200$ | 0.38 ± 0.48 | 13.0 | 10000 |
| SMC | $p = 25$ | 0.2 ± 0.4 | 3.21 | 2000 |
| | $p = 10$ | 0.28 ± 0.44 | 3.61 | 2000 |
| | $p = 5$ | 0.32 ± 0.46 | 4.58 | 2000 |
| | $p = 2$ | 0.39 ± 0.48 | 6.58 | 2000 |
| | $p = 1$ | 0.36 ± 0.48 | 10.31 | 2000 |
| BS | $p = 4$ | 0.25 ± 0.43 | 2.69 | 1020 |
| | $p = 3$ | 0.32 ± 0.46 | 4.11 | 1180 |
| | $p = 2$ | 0.39 ± 0.48 | 5.78 | 1500 |
| | $p = 1$ | 0.61 ± 0.48 | 10.79 | 2460 |
| SoP | $\Delta f = 40, \Delta b = 20$ | 0.25 ± 0.43 | 4.05 | 2940 |
| | $\Delta f = 20, \Delta b = 10$ | 0.54 ± 0.49 | 4.98 | 3420 |
| | $\Delta f = 10, \Delta b = 5$ | 0.76 ± 0.42 | 6.35 | 3860 |
| | $\Delta f = 8, \Delta b = 4$ | 0.85 ± 0.35 | 6.69 | 3900 |
| | $\Delta f = 6, \Delta b = 3$ | 0.91 ± 0.28 | 7.08 | 3940 |
| | $\Delta f = 4, \Delta b = 2$ | 0.95 ± 0.21 | 8.43 | 4220 |
| | $\Delta f = 2, \Delta b = 1$ | 0.98 ± 0.13 | 12.42 | 4980 |
| ABCD | percentage = 0.1, $\kappa = 1$ | 0.17 ± 0.37 | 0.65 | 310 |
| | percentage = 0.2, $\kappa = 1$ | 0.44 ± 0.49 | 1.22 | 562 |
| | percentage = 0.3, $\kappa = 1$ | 0.96 ± 0.19 | 2.75 | 1226 |
| | percentage = 1.0, $\kappa = 1$ | 0.99 ± 0.09 | 3.11 | 1400 |
| | percentage = 1.0, $\kappa = 3$ | 1.0 ± 0.0 | 3.75 | 1854 |

### B.11.2 Investigation on automatic thinking assignment across different difficulty levels

The Figure 15 visualize how our method allocates varying amounts of inference iterations and time depending on task difficulty for Pixel Maze. It demonstrates that, similar to Sudoku, the model adaptively assigns greater computational effort to larger and more complex mazes, confirming the generality of adaptive inference-time allocation across domains.

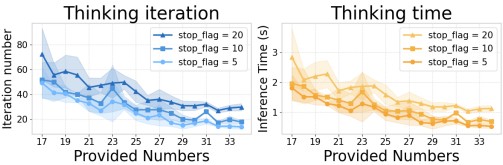 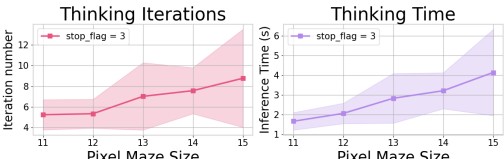

**Figure 14: Thinking iterations and inference time in Sudoku. Left:** Mean and standard deviation of iteration counts across varying numbers of provided digits. **Right:** Mean and standard deviation of per-instance inference time. The model adapts computation dynamically, allocating more iterations to harder puzzles with fewer initial clues.

**Figure 15: Thinking iterations and inference time in Pixel Maze. Left:** Mean and standard deviation of iteration counts across different maze sizes. **Right:** Mean and standard deviation of per-sample inference time. Larger mazes require more search, and the model allocates compute accordingly, demonstrating instance-aware inference adaptation.

### B.12 Molecular 3D structure prediction task experiment detail

Experiments were conducted on 100 molecules randomly sampled from the QM9 test dataset. For all methods, we utilized the same checkpoint of diffusion model from the official EDM repository, where $T = 1000$. Following [11], we compared molecule stability (the proportion of generated molecules for which all atoms are stable). We also measured the average inference time required for a single sample.

To compare inference-time scaling strategies, each baseline was scaled along its primary computational axis while preserving its core algorithmic behavior. **Base Diffusion** employed a single particle ($N = 1$). **BoN** was evaluated with using more $N$ to increase inference cost. **BS** and **SoP** were implemented with $M = 4$ and $K = 10$, and inference-time compute was scaled by increasing the branching and selection frequency. Specifically, for **BS**, we scale inference-time compute by adjusting the branching-and-selection frequency $p$. **SoP** additionally scaled computation by reducing the step sizes for back-and-forth moves $\Delta f$ and $\Delta b$. **SMC** used $N = 40$ particles and increased inference-time compute by adjusting the resampling frequency $p$ over different steps. Following the FKD [24], we use POTENTIAL TYPE = "max" and $\lambda = 100$ to make it as optimization. **ABCD (ours)** was configured with $N = 10$, $K = 10$, and $J = 5$, corresponding to four non-zero go-back noise levels (50, 100, 200, 400) and a zero level. The adaptive terminal condition halted inference once a specified fraction of top-$N$ particles repeatedly originated from the zero noise level for $\kappa$ consecutive steps. We scaled ABCD's inference time by varying both $\kappa \in \{1, 2\}$ and percentage thresholds $\in \{0.4, 0.5\}$. To ensure computational efficiency, ABCD employed 100 denoising steps, each executing a 10-step jump in the diffusion process.

## B.13 Molecular 3D structure prediction task experiment result

We provide detailed results corresponding to the main Figure 4(Right), with a breakdown of the scaling factors we used in Table 7.

**Table 7:** Comparison of various inference-time strategies across compute and performance in the QM9 setting.

| Inference Method | Add. Compute | Success rate | Clock Time (sec.) | NFE |
|---|---|---|---|---|
| Base Diffusion | No add | 0.74 | 4.07 | 1001 |
| BoN | $N = 40$ | 0.86 | 17.33 | 40040 |
| | $N = 80$ | 0.86 | 35.58 | 80080 |
| | $N = 160$ | 0.89 | 70.28 | 160160 |
| SMC | $p = 1$ | 0.88 | 24.17 | 48040 |
| | $p = 2$ | 0.85 | 35.18 | 60040 |
| | $p = 5$ | 0.9 | 52.29 | 80010 |
| BS | $p = 1$ | 0.82 | 24.17 | 24010 |
| | $p = 2$ | 0.83 | 37.95 | 45010 |
| | $p = 5$ | 0.91 | 69.95 | 80010 |
| SoP | $\Delta f = 50, \Delta b = 25$ | 0.9 | 32.21 | 73690 |
| | $\Delta f = 100, \Delta b = 50$ | 0.92 | 40.06 | 74410 |
| | $\Delta f = 200, \Delta b = 100$ | 94 | 49.84 | 75850 |
| ABCD | percentage = 0.4, $\kappa = 1$ | 0.93 | 4.35 | 7482 |
| | percentage = 0.4, $\kappa = 2$ | 0.97 | 23.09 | 43009 |
| | percentage = 0.5, $\kappa = 2$ | 0.99 | 46.45 | 82010 |

## B.14 Text-to-image generation experiment detail

We adopt Stable Diffusion v1.5, a widely used text-to-image diffusion model, as our pre-trained model. we comprehensively assessed ABCD's performance using three metrics: compressibility, measured as negative JPEG file size (in kilobytes) after compression following [2]; aesthetic evaluation, computed using LAION's V2 Aesthetic Predictor [22], a linear MLP built upon CLIP embeddings, trained on over 400,000 human ratings; and human preference evaluation, based on the HPSv2 scorer [33], a CLIP model fine-tuned on 798,090 human-selected rankings across 433,760 image pairs. For compressibility and aesthetic evaluations, we used animal category prompts from [2] (e.g., Dog, Cat, Panda), whereas for human preference evaluation we used human instruction prompts from [33].

Each inference method is evaluated by generating 12 images. Inference time is calculated as the average duration to produce one image. To analyze inference-time scaling, each baseline method was scaled by adjusting its primary computational parameters without altering its fundamental algorithmic design. **Base Diffusion** utilized a single particle ($N = 1$). **BoN** scaled inference by incrementing particle count from $N = 2$ up to $N = 12$. **SoP** increased computational cost through smaller incremental steps during both forward and backward transitions, denoted by $(\Delta f, \Delta b)$. Our proposed **ABCD** method employed $N = 4, K = 1,$ and $J = 4$, translating to three non-zero noise levels $(1, 301, 621)$ for backward transitions and one zero-noise level. The adaptive stopping criterion halted inference when the top-K particles persistently emerged from the zero noise level for $\kappa = 30$ consecutive steps. ABCD's inference scaling was achieved by varying the number of `max_iter`. For fast denoising, ABCD employed 50 inference steps with each step covering a 20-step jump in the diffusion space. Detailed results are provided in the next section B.15.

### B.15 Text-to-image generation experiment result

We present detailed results related to Figure 7, along with a breakdown of the scaling factors listed in Table 8, Table 9 and Table 10. For each method, inference-time compute was scaled by adjusting its primary control axis while preserving its core algorithmic behavior.

**Table 8:** Comparison of various inference-time strategies across compute and image compressibility in the Text-to-image generation task.

| Inference Method | Add. Compute | Compressibility | Clock Time (sec.) | NFEs |
|---|---|---|---|---|
| Base Diffusion | No add | $-100.07 \pm 2.88$ | 58.42 | 2000 |
| BoN | $N = 2$ | $-91.59 \pm 4.43$ | 109.25 | 4000 |
| | $N = 3$ | $-83.70 \pm 7.60$ | 150.17 | 6000 |
| | $N = 4$ | $-81.89 \pm 5.71$ | 200.00 | 8000 |
| | $N = 5$ | $-81.89 \pm 5.71$ | 257.58 | 10000 |
| | $N = 6$ | $-81.64 \pm 5.92$ | 292.75 | 12000 |
| | $N = 7$ | $-78.21 \pm 8.68$ | 348.33 | 14000 |
| | $N = 8$ | $-78.22 \pm 8.69$ | 403.00 | 16000 |
| | $N = 9$ | $-78.00 \pm 8.40$ | 444.58 | 18000 |
| | $N = 10$ | $-77.86 \pm 8.42$ | 494.33 | 20000 |
| | $N = 11$ | $-77.53 \pm 8.77$ | 556.33 | 22000 |
| | $N = 12$ | $-77.02 \pm 8.88$ | 583.25 | 24000 |
| SoP | $\Delta f = 510, \Delta b = 255$ | $-88.36 \pm 4.88$ | 188.75 | 7548 |
| | $\Delta f = 410, \Delta b = 205$ | $-85.32 \pm 3.21$ | 215.50 | 8490 |
| | $\Delta f = 350, \Delta b = 175$ | $-80.82 \pm 4.14$ | 227.92 | 9192 |
| | $\Delta f = 280, \Delta b = 140$ | $-72.64 \pm 5.61$ | 257.67 | 10266 |
| | $\Delta f = 220, \Delta b = 110$ | $-67.79 \pm 4.65$ | 271.25 | 10710 |
| | $\Delta f = 180, \Delta b = 90$ | $-64.59 \pm 3.43$ | 280.25 | 10974 |
| ABCD | `max_iter` $= 10$ | $-74.91 \pm 5.58$ | 41.83 | 1380 |
| | `max_iter` $= 20$ | $-67.19 \pm 5.84$ | 73.67 | 2360 |
| | `max_iter` $= 30$ | $-62.33 \pm 4.33$ | 103.67 | 3340 |
| | `max_iter` $= 40$ | $-59.01 \pm 4.34$ | 130.50 | 4320 |
| | `max_iter` $= 50$ | $-56.08 \pm 4.02$ | 163.67 | 5300 |
| | `max_iter` $= 80$ | $-51.49 \pm 4.34$ | 254.50 | 8240 |
| | `max_iter` $= 100$ | $-49.17 \pm 4.29$ | 306.08 | 10176 |

**Table 9:** Comparison of various inference-time strategies across compute and aesthetic score in the Text-to-image generation task.

| Inference Method | Add. Compute | Image Aesthetic | Clock time (sec.) | NFEs |
|---|---|---|---|---|
| Base Diffusion | No add | 5.57 ± 0.03 | 58.83 | 2000 |
| BoN | N = 2 | 5.63 ± 0.08 | 109.75 | 4000 |
| | N = 3 | 5.72 ± 0.09 | 150.92 | 6000 |
| | N = 4 | 5.75 ± 0.07 | 200.58 | 8000 |
| | N = 5 | 5.79 ± 0.05 | 258.33 | 10000 |
| | N = 6 | 5.81 ± 0.02 | 293.42 | 12000 |
| | N = 7 | 5.81 ± 0.02 | 349.00 | 14000 |
| | N = 8 | 5.82 ± 0.03 | 403.50 | 16000 |
| | N = 9 | 5.86 ± 0.05 | 445.25 | 18000 |
| | N = 10 | 5.86 ± 0.05 | 494.25 | 20000 |
| | N = 11 | 5.87 ± 0.04 | 556.75 | 22000 |
| | N = 12 | 5.90 ± 0.05 | 583.17 | 24000 |
| SoP | $\Delta f = 510, \Delta b = 255$ | 5.68 ± 0.03 | 189.42 | 7548 |
| | $\Delta f = 400, \Delta b = 200$ | 5.78 ± 0.03 | 206.83 | 9636 |
| | $\Delta f = 280, \Delta b = 140$ | 5.84 ± 0.08 | 258.67 | 10266 |
| | $\Delta f = 170, \Delta b = 85$ | 5.87 ± 0.04 | 285.08 | 10704 |
| | $\Delta f = 130, \Delta b = 65$ | 5.93 ± 0.08 | 301.42 | 11232 |
| | $\Delta f = 110, \Delta b = 55$ | 5.98 ± 0.02 | 313.67 | 11478 |
| ABCD | `max_iter` = 10 | 5.95 ± 0.06 | 40.00 | 1380 |
| | `max_iter` = 20 | 6.08 ± 0.06 | 72.17 | 2360 |
| | `max_iter` = 30 | 6.13 ± 0.09 | 105.42 | 3340 |
| | `max_iter` = 40 | 6.15 ± 0.08 | 132.00 | 4254 |
| | `max_iter` = 60 | 6.18 ± 0.05 | 178.75 | 6043 |
| | `max_iter` = 80 | 6.23 ± 0.07 | 233.83 | 7677 |
| | `max_iter` = 100 | 6.25 ± 0.09 | 288.42 | 9147 |

**Table 10:** Comparison of various inference-time strategies across compute and human preference score in the Text-to-image generation task.

| Inference Method | Add. Compute | Human Preference | Clock time (sec.) | NFEs |
|---|---|---|---|---|
| Base Diffusion | No add | 0.2710 ± 0.0028 | 61.00 | 2000 |
| BoN | N = 2 | 0.2749 ± 0.0013 | 112.25 | 4000 |
| | N = 3 | 0.2752 ± 0.0017 | 153.00 | 6000 |
| | N = 4 | 0.2752 ± 0.0018 | 203.08 | 8000 |
| | N = 5 | 0.2753 ± 0.0017 | 260.92 | 10000 |
| SoP | $\Delta f = 510, \Delta b = 255$ | 0.2725 ± 0.0031 | 192.83 | 7548 |
| | $\Delta f = 410, \Delta b = 205$ | 0.2728 ± 0.0030 | 221.00 | 8490 |
| | $\Delta f = 350, \Delta b = 175$ | 0.2739 ± 0.0044 | 233.00 | 9192 |
| | $\Delta f = 230, \Delta b = 115$ | 0.2751 ± 0.0029 | 272.25 | 10218 |
| | $\Delta f = 130, \Delta b = 65$ | 0.2765 ± 0.0046 | 811.25 | 11232 |
| | $\Delta f = 70, \Delta b = 35$ | 0.2777 ± 0.0039 | 335.42 | 11778 |
| | $\Delta f = 20, \Delta b = 10$ | 0.2787 ± 0.0027 | 431.42 | 13056 |
| ABCD | max_iter = 10 | 0.2776 ± 0.0027 | 46.83 | 1380 |
| | max_iter = 20 | 0.2791 ± 0.0021 | 82.83 | 2360 |
| | max_iter = 30 | 0.2797 ± 0.0023 | 115.92 | 3340 |
| | max_iter = 40 | 0.2804 ± 0.0027 | 142.17 | 4320 |
| | max_iter = 50 | 0.2808 ± 0.0028 | 183.42 | 5300 |
| | max_iter = 80 | 0.2815 ± 0.0030 | 272.08 | 8060 |
| | max_iter = 100 | 0.2819 ± 0.0032 | 322.08 | 9718 |
| | max_iter = 200 | 0.2823 ± 0.0033 | 432.92 | 12716 |

### B.15.1 Visualization of generated images

In this section, we show more generated samples. Figures 16, 17, and 18 compare outputs from baseline approaches and ABCD with respect to compressibility, aesthetic score, and human preference score, respectively.

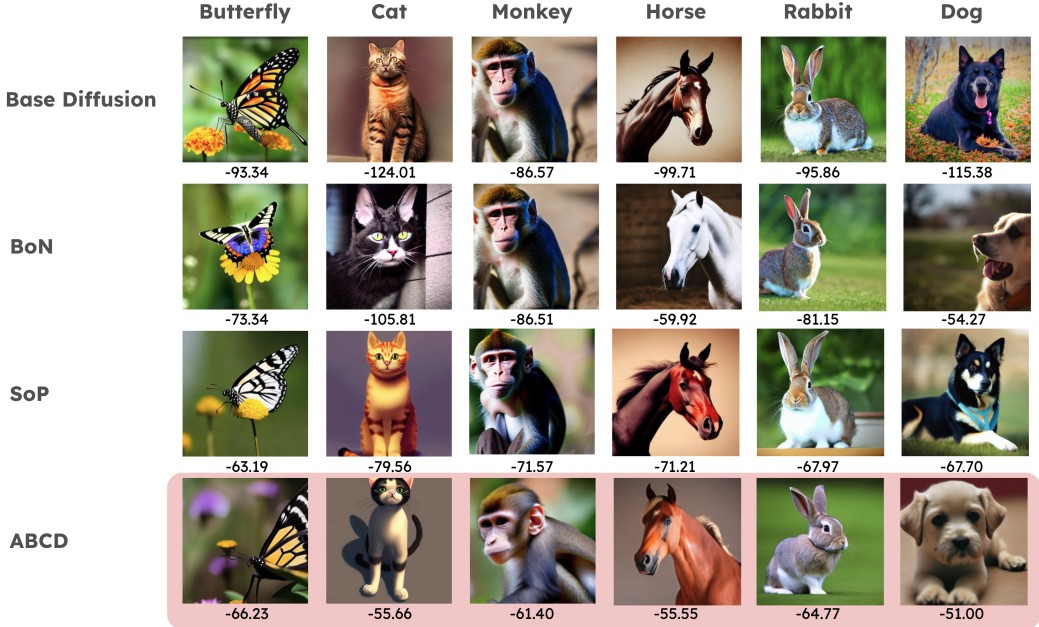

**Figure 16: Visual comparison of images produced by different inference-time strategies optimizing compressibility reward.** Numbers at the bottom of each image show the corresponding compressibility scores. **Base Diffusion** produces photorealistic textures and complex backgrounds, but at the cost of very large file sizes. **BoN** occasionally yields simpler backgrounds but still leaves a lot of high-frequency detail. **SoP** further flatten color gradients, shaving further off file size. Under **ABCD**, each image sports a clean, uncluttered background and smoother regions—e.g., the Cat is rendered with minimal fur grain, the Horse with a crisp silhouette against a solid tone.

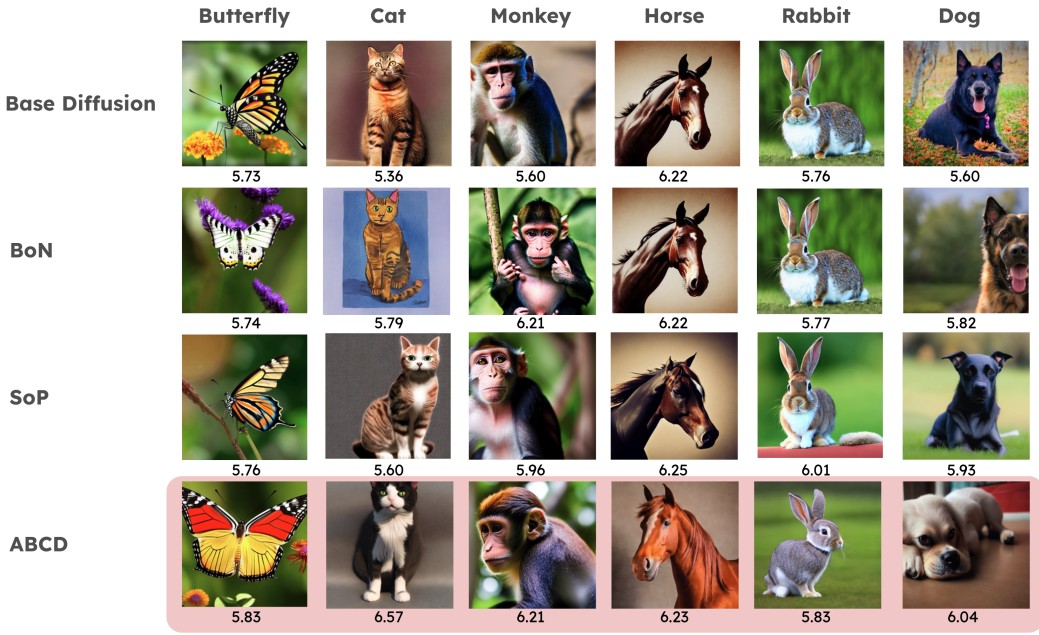

**Figure 17: Visual comparison of images produced by different inference-time strategies optimizing aesthetic score reward.** Numbers at the bottom of each image show the corresponding aesthetic scores.

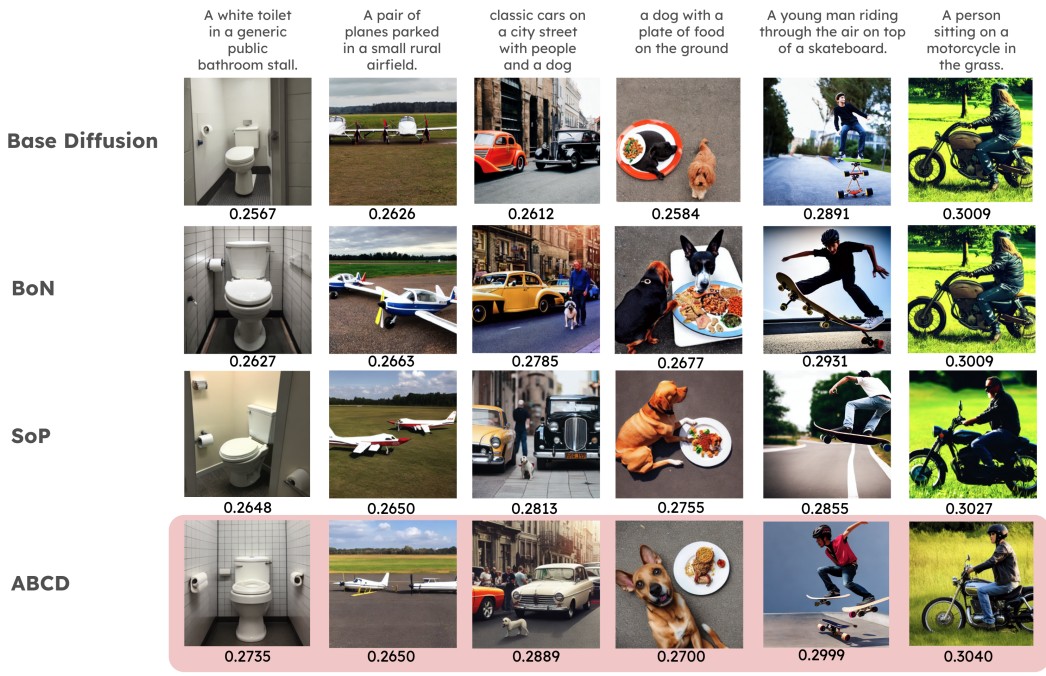

**Figure 18: Visual comparison of images produced by different inference-time strategies optimizing human preference score reward.** Numbers at the bottom of each image show the corresponding human preference scores.

## B.16 Comparison with respect to number of function evaluations of the diffusion model

In this section, we present a more detailed and fair comparison with baselines by evaluating not only the wall-clock time but also the Number of Function Evaluations (NFEs) during inference across tasks. The NFE measures how many times the pretrained diffusion function is called, and to isolate pure computational cost from parallelization effects, we count one function evaluation per particle per step. As shown in Figure 19 20, ABCD demonstrates superior inference-time scaling compared to other baselines, consistent with wall-clock time results.

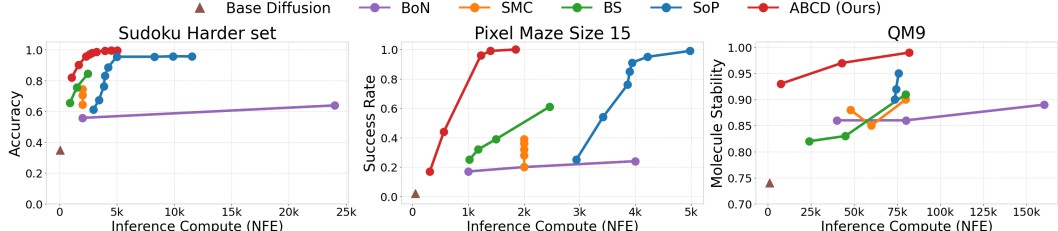

**Figure 19: Comparison w.r.t NFEs. Left:** Mean accuracy on Sudoku Harder dataset (17-28 entities provided). **Middle:** Pixel maze path finding result. (Size 15). Success rate on the OOD size-15 pixel maze test set. **Right:** Molecular 3D structure prediction task result. Molecular Stability rate on the QM9 dataset.

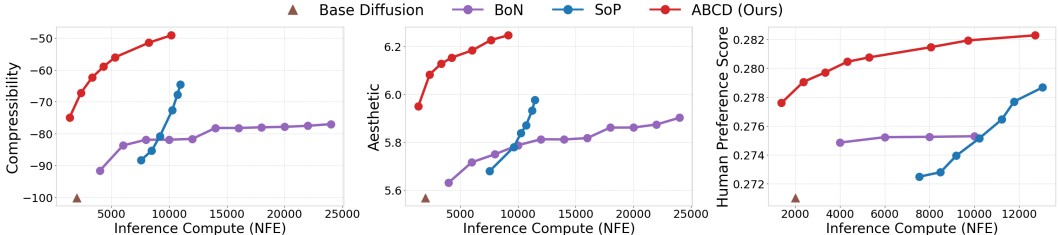

**Figure 20: Comparison w.r.t NFEs on Text-to-Image generation task.** Text-to-image generation result. Average reward with respect to compressibility, aesthetic score and human preference score.

# C  Ablation studies

## C.1  Investigate on existance of inference time scalable *go-back* noise level per task

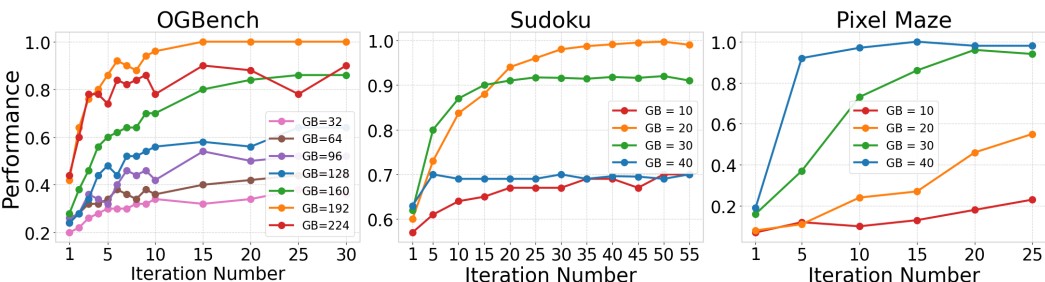

**Figure 21: Existence of Inference-Time Scalable *go-back* Noise Level per Task. Left:** OGBench Point Maze Giant. **Middle:** Sudoku harder dataset(17 to 34 provided). **Right:** Pixel Maze (Size 15). Performance across iterations for different *go-back* noise levels. Each task exhibits a distinct optimal *go-back* level, demonstrating the necessity of task-specific noise-level selection for effective inference-time scaling.

**Existence of inference time scalable *go-back* noise level per tasks.** To investigate the impact of the *go-back* noise level on inference-time scaling, we conducted an additional study across OGBench Point Maze, Sudoku, and Pixel Maze. As shown in Figure 21, the optimal *go-back* noise level—expressed as a fraction of the total denoising steps—varies significantly across tasks: approximately $3/4$ for OGBench, $2/5$ for Sudoku, and $4/5$ for Pixel Maze. These results highlight that selecting an appropriate *go-back* step size is critical for effective iterative refinement during inference time.

## C.2  Investigate on different configuration of Adaptive Thinking Time

**As the terminal condition becomes more stringent, both performance and inference cost increase.** To better understand how different computational budgets affect final performance, we added ablation study analyzing the core components of ABCD—Adaptive Thinking Time and Automatic Exploration-Exploitation Balancing—under varying configurations. Specifically, we conducted experiments that vary the level of computation allocated to each mechanism to examine how efficiency and accuracy are impacted in both the Sudoku and text-to-image generation domains.

In the Sudoku task, We focused on the Adaptive Thinking Time mechanism by varying the termination percentage and the value of $\kappa$, two key hyperparameters that define ABCD's adaptive terminal condition. As shown in the two tables 11, 12, increasing $\kappa$ generally improves accuracy but also leads to longer inference time, and similarly, raising the termination percentage improves performance at the cost of additional computation. These results show that by adjusting the strictness of the termination criterion, we can effectively control the trade-off between accuracy and efficiency.

Table 11: ABCD – Sudoku (Changing $\kappa$). We report accuracy, wall-clock time, and NFEs under different $\kappa$ values.

| Metric | $\kappa = 1$ | $\kappa = 5$ | $\kappa = 10$ |
|---|---|---|---|
| Accuracy | 0.958 | 0.986 | 0.994 |
| Time (s) | 0.443 | 0.608 | 0.754 |
| NFEs | 2296 | 3205 | 3942 |

Table 12: ABCD – Sudoku (Changing percentage). We report accuracy, wall-clock time, and NFEs under different percentage values.

| Metric | perc=0.4 | perc=0.6 | perc=0.8 |
|---|---|---|---|
| Accuracy | 0.698 | 0.821 | 0.903 |
| Time (s) | 0.112 | 0.210 | 0.329 |
| NFEs | 520 | 1059 | 1667 |

In the text-to-image generation task, we analyzed the Automatic Exploration–Exploitation Balancing mechanism by varying the granularity of the temperature pool—specifically, the number of go-back temperatures included in the pool. We fixed the number of particles and terminal conditions (max_iter=50, $\kappa = 30$) and varied the temperature pool size. As shown in the two tables 13, 14, using a finer pool (i.e., more diverse temperature values) improves the final compressibility and aesthetic score but also increases both inference time and NFE. This experiment highlights the impact of temperature diversity on controllability and performance.

**Table 13: More diverse temperature values on Compressibility score.** We report Compressibility score, wall-clock time, and NFEs under different temperature pool size.

| $\mathcal{T}$ **size** | Comp | Time (s) | NFEs |
|---|---|---|---|
| 2 | -55.1±4.9 | 233.3 | 8188 |
| 3 | -48.8±2.1 | 285.8 | 10500 |
| 4 | -46.1±2.1 | 356.5 | 12800 |

**Table 14: More diverse temperature values on Aesthetic score.** We report Aesthetic score, wall-clock time, and NFEs under different temperature pool size.

| $\mathcal{T}$ **size** | Aesth | Time (s) | NFEs |
|---|---|---|---|
| 2 | 6.23±0.05 | 237.9 | 8200 |
| 3 | 6.31±0.04 | 286.5 | 10369 |
| 4 | 6.36±0.14 | 354.6 | 12578 |

## C.3 Comparison with more intelligent *go-back* strategy

Since some amount of unnecessary compute is being performed by distributing each sample uniformly along the noise levels, we explored a more intelligent go-back strategy—Adaptive ABCD—but found it doesn't show meaningful improvement. Specifically, we tested strategy where particles were redistributed to different noise levels in proportion to the average reward observed at those levels across the population. However, as shown in the table 15, despite its intuitive appeal, the performance improvement was marginal compared to the added complexity. Therefore, we ultimately chose a simpler and more efficient formulation for the current version of ABCD. Nevertheless, we believe that incorporating smarter strategies—such as applying ideas from SMC to the temperature pool—could be a promising direction.

**Table 15:** Comparison between Adaptive ABCD and ABCD. We report accuracy and wall-clock time under different compute configurations.

| Method | Add. Compute | Accuracy | Wall Clock Time (s) |
|---|---|---|---|
| **Adaptive ABCD** | perc=0.6, $\kappa$=1 | 0.810 | 0.220 |
| | perc=0.8, $\kappa$=1 | 0.895 | 0.395 |
| | perc=1.0, $\kappa$=1 | 0.951 | 0.489 |
| | perc=1.0, $\kappa$=3 | 0.979 | 0.549 |
| **ABCD** | perc=0.6, $\kappa$=1 | 0.821 | 0.240 |
| | perc=0.8, $\kappa$=1 | 0.903 | 0.350 |
| | perc=1.0, $\kappa$=1 | 0.958 | 0.450 |
| | perc=1.0, $\kappa$=3 | 0.979 | 0.557 |

## C.4 Investigate on the diversity preservation of ABCD

We measured diversity using the average pairwise cosine similarity of CLIP embeddings, where a lower score reflects greater variability in outputs and broader exploration of the data space. While optimization naturally reduces some diversity, our method successfully preserves the underlying multi-modality of the pretrained model while steering toward high-reward regions.

**Table 16: Comparison of diversity scores across Compressibility, Aesthetic, and HPS tasks.** We report the diversity mean $\pm$ std under different compute configurations.

| Method | Add compute | Compressibility | Aesthetic | HPS |
|---|---|---|---|---|
| Base Diffusion | - | 0.2957±0.0481 | 0.2957±0.0481 | 0.4559±0.0215 |
| BoN | N=5 | 0.2859±0.0190 | 0.2870±0.0102 | 0.4424±0.0095 |
| | N=10 | 0.2781±0.0184 | 0.2919±0.0285 | - |
| SoP | $\Delta f$=510, $\Delta b$=255 | 0.2795±0.0375 | 0.2912±0.0308 | 0.4677±0.0227 |
| | $\Delta f$=280, $\Delta b$=140 | 0.2975±0.0512 | 0.3005±0.0199 | - |
| | $\Delta f$=230, $\Delta b$=115 | - | - | 0.4508±0.0152 |
| | $\Delta f$=180, $\Delta b$=90 | 0.2798±0.0154 | - | - |
| | $\Delta f$=170, $\Delta b$=85 | - | 0.2672±0.0323 | - |
| | $\Delta f$=130, $\Delta b$=65 | - | - | 0.4417±0.0054 |
| ABCD | max_iter=30 | 0.3128±0.0084 | 0.2848±0.0221 | **0.4672±0.0160** |
| | max_iter=80 | **0.3162±0.0028** | **0.2888±0.0222** | 0.4543±0.0143 |
| | max_iter=100 | 0.3037±0.0171 | 0.2880±0.0242 | 0.4555±0.0167 |

# D  Prediction visualization

In this section, we provide illustrative visualizations of instance-level predictions generated by our model to facilitate intuitive understanding. Figure 22 displays representative outputs from both the Pixel Maze and Sudoku tasks.

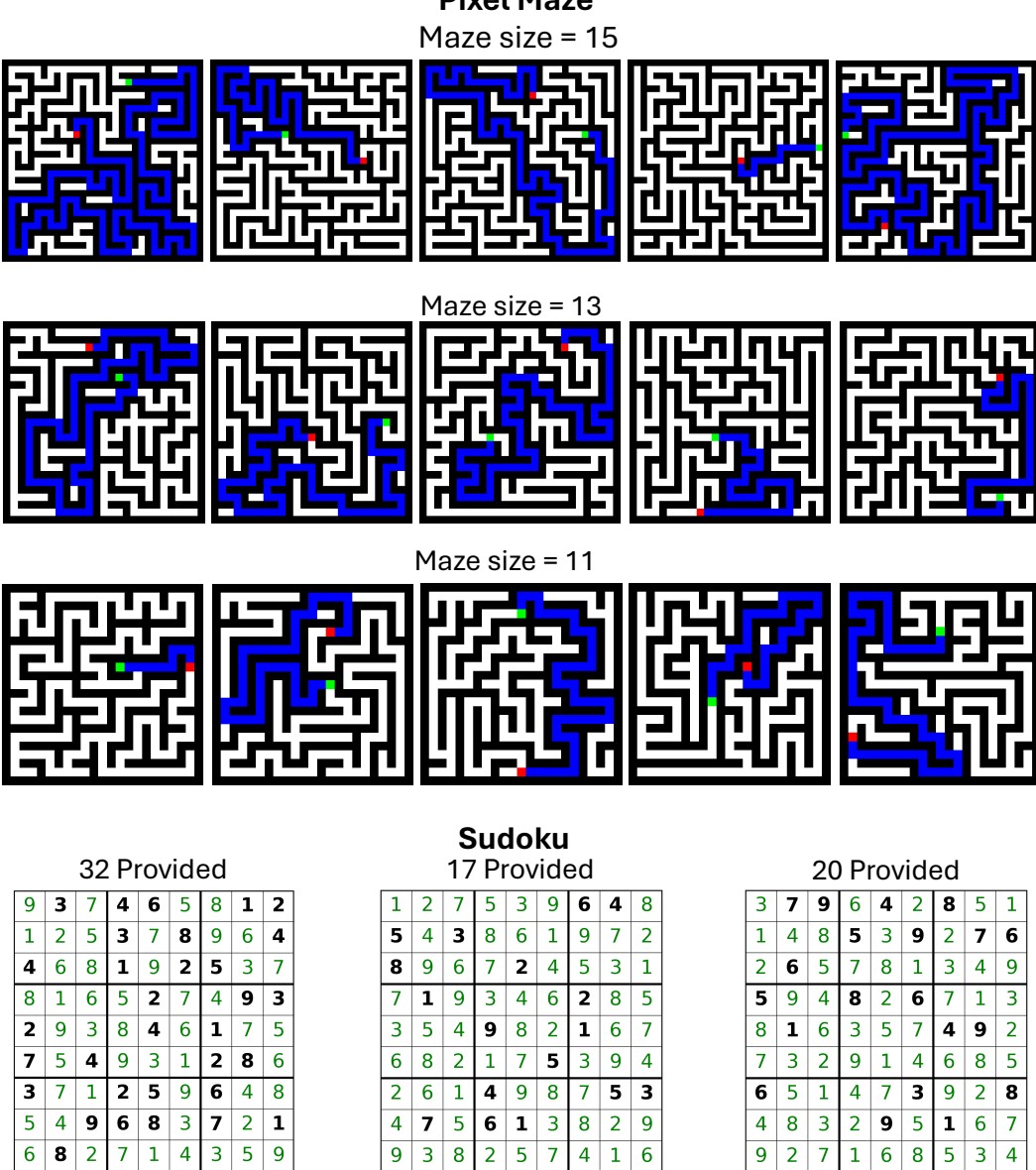

**Figure 22:** This figure presents the final outputs obtained by our ABCD method on both the Pixel Maze and Sudoku tasks. The visualizations demonstrate that, across a range of maze sizes and Sudoku difficulty levels, ABCD produces accurate predictions, particularly in instances that demand greater computational effort.

# E    Termination guarantee of the Adaptive Thinking Time

In this section, we present a detailed proof establishing the termination guarantee of the Adaptive Thinking Time algorithm introduced in Section 3.3.

## E.1    Formal statement

Let
$$T = \{t_1, \ldots, t_m\}$$
be the "temperature pool" with lowest temperature $t_1 = 0$. At cycle $c$, let $D_k(c)$ be the multiset of origin-temperatures of the top-$k$ particles after denoising. Define the stopping rule: terminate at the first cycle $N$ such that

$$\forall i \in \{N - \kappa + 1, \ldots, N - 1, N\}, \quad D_k(i) = \{0, 0, \ldots, 0\},$$

i.e. all top-$k$ particles originate from $t_1 = 0$ for $\kappa$ consecutive cycles.

**Assumptions**

1. **Bounded reward.** There is a reward function $r : \mathbb{R}^d \to \mathbb{R}$ with finite supremum $r^* < \infty$.

2. **Monotonic selection.** Let
$$r_k(c) = \min\{\text{rewards of top-}k \text{ at cycle } c\}.$$
   Then $r_k(c)$ is non-decreasing in $c$.

3. **Attainment.** There exists at least one state $x^*$ such that $r(x^*) = r^*$.

**Theorem 1.** *Under these assumptions, ABCD's Adaptive Thinking Time criterion triggers termination in finite time.*

The proof of this theorem is built upon several key lemmas, which we will now introduce and prove.

## E.2    Uniform Hit-Chance

**Lemma E.1.** *Let $x^*$ be any maximizer of the reward, $r(x^*) = r^*$. Under the ABCD dynamics (Algorithm 1), there is a constant $p > 0$ such that for each cycle $c$,*
$$\Pr\big(E_c \mid \mathcal{F}_{c-1}\big) \geq p,$$
*where $E_c = $ "$x^*$ appears among the selected top-K at cycle $c$" and $\mathcal{F}_{c-1}$ is the $\sigma$-algebra of all randomness up to (but not including) cycle $c$.*

*Proof.*    1. At the start of cycle $c$ we have $K$ anchor particles at $t = 0$. We replicate each $J$ times and send each of those $KJ$ copies through the forward noising process $q(x_{t'} \mid x_0)$ at every $t' \in T$. Exactly $K$ of those replicas go to the highest temperature $t_m$.

2. Conditioned on $\mathcal{F}_{c-1}$, the noising and denoising of these $KJ$ particles is independent of the past and identically distributed across cycles. In particular, the $K$ samples at $t_m$ are fresh each time. Let
$$q = \Pr\big[(\text{noising at } t_m) \text{ then (denoise) yields } x^*\big].$$
   By full support of the forward kernel and nonzero recovery probability of the denoiser, $q > 0$.

3. Those $K$ draws at $t_m$ are independent, so the chance none equals $x^*$ is $(1 - q)^K$. Hence
$$p = 1 - (1 - q)^K > 0.$$

4. If any replicate equals $x^*$, its reward is $r^*$, guaranteeing $x^*$ is in the top-$K$. Thus for every history,
$$\Pr\big(E_c \mid \mathcal{F}_{c-1}\big) = \Pr\big(\text{some } t_m\text{–replicate} = x^* \mid \mathcal{F}_{c-1}\big) \geq p.$$

This completes the proof of the uniform hit-chance lemma.    □

### E.3 Optimality

**Lemma E.2.** *Under the assumptions*

1. Monotonicity and Boundedness: $r_k(c)$ *is non-decreasing and* $r_k(c) \leq r^* < \infty$, *so* $r_k(c) \to r_\infty \leq r^*$.

2. Supremum Attained: $\exists\, x^*$ *with* $r(x^*) = r^*$.

3. Uniform Hit-Chance: *at each cycle c,* $\Pr(E_c \mid past) \geq p > 0$.

*Then*
$$\Pr\big[r_\infty = r^*\big] \;=\; 1.$$

*Proof.*   1. Define the "hit" event $E_c$ as above. By assumption 3, $\sum_{c=1}^{\infty} \Pr(E_c \mid \text{past}) \geq \sum_c p = \infty$.

2. By the conditional Borel–Cantelli lemma [23], $\Pr(E_c \text{ i.o.}) = 1$, so infinitely many hits occur almost surely.

3. Let $C$ be the first cycle where $E_C$ occurs. Then
$$r_k(C) \geq r(x^*) = r^*, \quad r_k(C) \leq r^*, \quad \implies \quad r_k(C) = r^*.$$

4. Monotonicity then implies $r_k(c) = r^*$ for all $c \geq C$, so $r_\infty = r^*$.

Hence $\Pr[r_\infty = r^*] = 1$. □

### E.4 Finite-Time Hitting

**Lemma E.3.** *Under the* Uniform Hit-Chance *lemma, define*
$$T \;=\; \inf\big\{\, c \geq 1 \mid x^* \text{ appears among the top-K at cycle } c\big\}.$$

*Then*
$$\Pr\big(T < \infty\big) = 1, \quad \Pr\big(T > n\big) \leq (1-p)^n \quad \forall n \in \mathbb{N},$$

*where* $p > 0$ *is the per-cycle success lower bound.*

*Proof.*   1. **Failure-to-hit bound.**
Let
$$F_n \;=\; \bigcap_{c=1}^{n} E_c^c, \quad E_c = \{\text{cycle } c \text{ hits } x^*\}.$$

By the Uniform Hit-Chance lemma,
$$\Pr\big(E_c \mid \mathcal{F}_{c-1}\big) \geq p \quad \implies \quad \Pr\big(E_c^c \mid \mathcal{F}_{c-1}\big) \leq 1 - p.$$

Hence, by the tower property,
$$\Pr(F_n) = \Pr\big(E_1^c \cap \cdots \cap E_n^c\big) = \mathbb{E}\big[\Pr(E_n^c \mid \mathcal{F}_{n-1})\, \mathbf{1}_{F_{n-1}}\big] \leq (1-p)\, \Pr(F_{n-1}),$$

and by induction
$$\Pr(F_n) \leq (1-p)^n.$$

2. **Hitting in finite time w.p.1.**
Since $\{T > n\} = F_n$, we get
$$\Pr(T > n) \leq (1-p)^n \xrightarrow{n \to \infty} 0,$$

so $\Pr(T = \infty) = 0$ and hence $\Pr(T < \infty) = 1$.

□

### E.5 Proof of Main Theorem

We now assemble the results from the lemmas to prove Theorem 1.

1. **Monotone convergence of $r_k(c)$.** The sequence $r_k(0), r_k(1), r_k(2), \ldots$ is non-decreasing and bounded above by $r^*$, hence it converges to some limit $r_\infty \leq r^*$.

2. **Limit must equal $r^*$.** According to Lemma E.2, $r_\infty = r^*$.

3. **Optimal is reached in finite time.** By Lemma E.3, $r_k(c)$ hits $r^*$ at some finite cycle $C$.

4. **Once optimal is reached, only $t_1$-origins survive.** At cycle $C$ where $r_k(C) = r^*$, any particle noised at temperatures $t > 0$ and then denoised back cannot exceed $r^*$. Thus the only way to preserve the top-$k$ set of reward-$r^*$ particles is via replicas from $t_1 = 0$, which leave them unchanged. Consequently, for all $c \geq C$, $D_k(c) = \{0, 0, \ldots, 0\}$.

5. **$\kappa$-persistence triggers termination.** Therefore, once cycle $C$ is reached, the condition "all top-$k$ from temperature 0 for $\kappa$ consecutive cycles" is satisfied by cycle $C + \kappa - 1$, and ABCD terminates.

