# OpenReview forum: "Adaptive Inference-Time Scaling via Cyclic Diffusion Search"
_NeurIPS.cc/2025/Conference — NeurIPS 2025 poster_

### Official Review · Reviewer_jBRb · 2025-06-25

**Clarity:** 4
**Significance:** 4
**Originality:** 2
**Rating:** 4
**Confidence:** 4

**Summary:**

This paper proposes an idea to use a pre-trained diffusion model to generate $\arg\max_x r(x)$ for some black-box reward function $r$. To simultaneously handle exploitation and exploration, they map the initially predicted data into multiple noise levels and re-denoise them in parallel. They also propose a way to adaptively terminate the sampling process.

**Questions:**

- Have the authors tried a simpler adaptive termination method that only checks whether the reward is improving or not? Does the proposed method offer any advantage over this simple baseline? The proposed method terminates when high-reward samples are mostly from small $t'$, but this discards the case where local exploration keeps improving, in which case it may be too early to terminate the algorithm.
- Intuitively, low $t'$ encourages exploitation, whereas a large $t'$ promotes exploration. Does this mean that in environments where exploration is more important (e.g., when the reward support is far from the pre-trained model's support), BoN will outperform the proposed method? I can imagine that when all initial samples receive very low rewards, adding small noises to them and denoising them again can be a waste of compute. It might be better to just start with fresh samples.
- As mentioned in the paper, SoP uses a similar back-and-forth denoising->re-noising process. What is the benefit of ABCD over SoP? One disadvantage of SoP (and all other baselines compared) is that it uses $r(\mathbb{E}[x_0 \mid x_t])$ as an approximation of the true reward, which is not very reliable--correct? Is there any advantage of SoP over ABCD? A more detailed comparison would be helpful.


>For fair comparison, the total number of particles was set equal across all methods (except BoN).

Does the same number of particles imply the same number of forward passes of the score function? If not, it would be better to report the latter as well, so that readers can clearly compare the computational efficiency in terms of FLOPs.

For the same reason, wouldn’t it be better to report the number of forward passes instead of “Mean Cyc” in Table 2?

In Table 1, what does “success” mean? The distance to the goal is not 0. Do you use a threshold value and consider a particle successful when the distance is below that threshold? This part is not clearly explained.

**Ethical Concerns:**

["NO or VERY MINOR ethics concerns only"]

**Final Justification:**

The proposed adaptive rule is equivalent to the naive algorithm that simply "checks whether the reward has stagnated." Terminating a search algorithm when a criterion stops improving is a general idea and not particularly new. The current explanation seems to be an overly complicated way to describe this simple algorithm. As this is presented as a core contribution of the paper, upon the author's clarification, I'm lowering my score.

**Limitations:**

yes

**Paper Formatting Concerns:**

Looks good

**Quality:**

4

**Strengths And Weaknesses:**

# Strengths:

- The idea is simple, and the motivation is sound. The concept of performing both exploitation and exploration in a parallel manner is interesting.
- The topic--learning to optimize test-time reward using a pre-trained generative prior--is timely and of interest to the community.

# Weaknesses:

- Experiments are done on a relatively small scale. However, this is sufficient to analyze the proposed method in a controlled way, so this is not considered a significant weakness in the reviewer's opinion.
- Always denoising all samples from all $t'$ seems inefficient. This could be made more efficient by selecting $t'$ more intelligently.
- The adaptive termination criterion could have been compared to a simpler stopping criterion that checks whether the reward has stagnated.
- In the experimental results, the performance gap between some existing baselines appears moderate. For instance, in Sudoku, SoP performs on par with the proposed method.
- Comparing wall-clock time does not seem to provide the full picture. See questions.

---

> ### Author Rebuttal · Authors · 2025-07-29
>
> We are grateful for your constructive feedback and your recognition of our efforts. In particular, we appreciate your insightful comments on **intelligently selecting the go-back temperature, the potential for continued improvement through small refinements, the intuitive BoN-based illustration of ABCD behavior, the comparison with SoP, and your careful attention to experimental settings**. Below, we provide responses to all of the questions and concerns you raised:
> ### **[Question 1 & Weakness 3] About the simpler termination method & premature termination issue**
> Thank you for the insightful question. As you pointed out, there are indeed situations where local exploration continues to improve. **To account for this, our ABCD framework always includes the lowest go-back temperature (i.e., 0) across all tasks.** In effect, the algorithm terminates only when the best sample is the one that does not require further refinement for some consecutive cycles. **Additionally, the issue of early termination can be mitigated by using a stricter $\kappa$ condition as well.**
> ### **[Question 2] I can imagine that when all initial samples receive very low rewards, adding small noises to them and denoising them again can be a waste of compute. It might be better to just start with fresh samples.**
> This is another very intuitive question. **One of the strengths of ABCD is that it can handle all these cases.** This is because it sends the particles across all temperature levels. In the scenario you mentioned, ABCD would likely choose the particle sent to the highest temperature $T$, allowing ABCD to effectively perform a full restart when necessary.
> ### **[Question 3] What is the benefit of ABCD over SoP?**
> We believe that ABCD offers several key advantages over SoP.
>
> 1. **First of all, ABCD enables unbounded cycling between denoising and re-noising, whereas SoP does not allow further refinement once it reaches $t = 0$**. This limits SoP’s ability to scale inference-time computation. In contrast, ABCD introduces an additional axis of scalability by allowing repeated refinement cycles even after reaching $t = 0$, enabling it to flexibly allocate more computation based on task complexity.
> 2. **ABCD adapts the generative process on a per-instance basis**. ABCD allows on-line adaptive inference-time compute whereas SoP has constant & freezed inference computation regardless of the difficulty or other requirements such as desired quality-speed tradeoff.
> 3. As you correctly mentioned, unlike SoP, **ABCD does not rely on approximating the reward using** $r(\mathbb{E}[x_0 | x_t])$. Instead, it evaluates the actual reward at the generated $x_0$ via fast denoising, thereby avoiding the approximation error inherent in SoP’s approach.
>
> These properties collectively make ABCD a more flexible and potentially more powerful framework for guided generation.
> ### **[Weakness 1] As pointed out by another reviewer, large-scale experiments are not strictly necessary, as evaluating the proposed method in a controlled setting—as done in the current work—is sufficient. Therefore, small-scale experimentation is not considered a Weakness.**
> **Thank you for advocating the value of our relatively small-scale experiments.** At the same time, several reviewers raised concerns about the scalability of our approach to larger domains such as images. **In response, we conducted additional experiments in the text-to-image domain. The results were consistent with our other experiments.** The results of which are discussed in **[Question 1] of reviewer WZvc**.
> ### **[Weakness 2] selecting $t' $ more intelligently can be possible**
> Thank you for pointing out this highly relevant issue—we completely share the motivation behind it. During the development of our method, **we explored a more intelligent alternative go-back strategy but found it doesn’t show meaningful improvement.**
>
> For instance, we also experimented with a more informed go-back mechanism, where particles were redistributed to different noise levels in proportion to the average rewards observed at those levels across the population. As shown in the results below, however, **this approach introduced additional complexity without yielding meaningful performance improvements.** Consequently, we chose to adopt a simpler version in the final design of ABCD.
>
> Nonetheless, as you insightfully suggested, incorporating more sophisticated techniques—such as leveraging concepts from SMC within the temperature pool—could further enhance the method. We consider this a compelling avenue for future exploration.
> - **Adaptive ABCD vs. ABCD in Sudoku**
> |Method|Add. Compute|Accuracy|Clock Time (s)|
> |-|-|-|-|
> |**Adaptive ABCD**|perc=0.6,𝜅=1|0.810|0.220|
> ||perc=0.8,𝜅=1|0.895|0.395|
> ||perc=1.0,𝜅=3|0.979|0.549|
> |**ABCD**|perc=0.6,𝜅=1|0.821|0.240|
> ||perc=0.8,𝜅=1|0.903|0.350|
> ||perc=1.0,𝜅=3|0.979|0.557|
> ### **[Question 4 & Weakness 5] Does the same number of particles imply the same number of forward passes of the score function? If not, it would be better to report the latter as well, so that readers can clearly compare the computational efficiency in terms of FLOPs. & [Weakness 4] In the experimental results, the performance gap between some existing baselines appears moderate. For instance, in Sudoku, SoP performs on par with the proposed method.**
> Indeed, the same number of particles does not necessarily imply an equal number of score function evaluations. **To clarify this, we explicitly report the Number of Function Evaluations (NFE) per 1 sample, which measures the total number of score function calls during inference. We found that evaluating in NFE provides a more effective way to highlight the advantage of our method over baselines.** **Specifically, in tasks like Sudoku, the performance and speed gap between ABCD and existing methods becomes more pronounced, effectively addressing your concern in [Weakness 4]**. Although we can not include the updated figure here, we provide the updated summary table below and will incorporate the full results in the final camera-ready version to ensure a clear and fair comparison.
> - **Sudoku**
> |Method|Config|Accuracy|NFEs|
> |-|-|-|-|
> |**Base Diffusion**|No add|0.35|50|
> |**Best-of-N (BoN)**|N=480|0.639|24000|
> ||N=960|0.662|48000|
> |**SMC**|p=5|0.643|2000|
> ||p=1|0.745|2000|
> |**Beam Search**|p=2|0.757|1500|
> ||p=1|0.844|2460|
> |**Search-over-Path (SoP)**|Δf=10, Δb=5|0.763|3860|
> ||Δf=4, Δb=2|0.886|4220|
> ||Δf=6, Δb=5|**0.957**|**11540**|
> |**ABCD (Ours)**|perc=0.6, 𝜅=1|0.821|1059|
> ||perc=1.0, 𝜅=5|0.986|3205|
> ||perc=1.0, 𝜅=20|**0.996**|**5013**|
> - **Pixel Maze**
> |Method|Config|Success Rate|NFEs|
> |-|-|-|-|
> |**Base Diffusion**|No add|0.02|50|
> |**Best-of-N (BoN)**|N=40|0.20|2000|
> ||N=160|0.34|8000|
> ||N=200|0.38|10000|
> |**SMC**|p=5|0.32|2000|
> ||p=1|0.36|2000|
> |**Beam Search**|p=2|0.39|1500|
> ||p=1|0.61|2460|
> |**Search-over-Path (SoP)**|Δf=10, Δb=5|0.76|3860|
> ||Δf=8, Δb=4|0.85|3900|
> ||Δf=4, Δb=2|**0.95**|**4220**|
> |**ABCD (Ours)**|perc=0.2, 𝜅=1|0.44|562|
> ||perc=0.3, 𝜅=1|0.96|1226|
> ||perc=1.0, 𝜅=3|**1.00**|**1854**|
> - **OGBench Maze Giant**
> |Method|Config|Performance|NFEs|
> |-|-|-|-|
> |**Base Diffusion**|No add|12±13|1280|
> |**Best-of-N (BoN)**|N=64|54±22|81920|
> ||N=288|64±17|368640|
> ||N=544|76±20|696320|
> |**SMC**|p=20|20±13|41376|
> ||p=10|22±11|41792|
> |**Beam Search**|p=5|50±16|132960|
> ||p=1|72±16|618560|
> |**Search-over-Path (SoP)**|Δf=100,Δb=50|60±15|59664|
> ||Δf=50,Δb=25|86±13|71784|
> ||Δf=6,Δb=3|**96±12**|**98360**|
> |**ABCD (Ours)**|max_iter=5|94±9|19020|
> ||max_iter=10|96±8|29720|
> ||max_iter=35|**100±0**|**83220**|
> - **OGBench Maze Large**
> |Method|Con|Performance|NFEs|
> |-|-|-|-|
> |**Base Diffusion**|No add|38±11|1280|
> |**Best-of-N (BoN)**|N=4|70±16|5120|
> ||N=8|80±9|10240|
> ||N=16|88±10|20480|
> |**SMC**|p=20|82±11|41376|
> ||p=10|76±15|41792|
> |**Beam Search**|p=15|84±12|52720|
> ||p=5|88±10|132960|
> |**Search-over-Path (SoP)**|Δf=100,Δb=50|76±12|58164|
> ||Δf=50,Δb=25|86±9|71034|
> ||Δf=6,Δb=3|**96±8**|**98270**|
> |**ABCD (Ours)**|J=32|**100±0**|**6360**|
> - **Image Generation**
>
> Please refer to our response to **[Question 1] from Reviewer WZvc**.
> ### **[Question 5] report the number of forward passes instead of “Mean Cyc” in Table 2?**
> The purpose of Table 2 is to highlight the necessity of employing multiple go-back step sizes. However, we agree that including the number of forward passes would provide a clearer picture of computational costs. **We add the NFEs to Table 2 and observe the same trend, further reinforce our conclusion**. We will update the table accordingly to reflect this clearly.
> |Method|Success rate|Mean Cyc|NFEs|Success rate|Mean Cyc|NFEs|
> |-|-|-|-|-|-|-|
> |**ABCD (Ours)**|**100**|**3.36**|**308.32**|**100**|**7.02**|**535.24**|
> |GB 21|**100**|**2.06**|**182.4**|79|16.14|745.6|
> |GB 41|8|46.38|3439.36|**100**|**6.77**|**587.44**|
> |GB 61|0|–|–|100|15.05|1665.2|
> |GB 81|0|–|–|6|43.33|5992.88|
> ### **[Question 6] In Table 1, what does “success” mean? The distance to the goal is not 0. Do you use a threshold value and consider a particle successful when the distance is below that threshold? This part is not clearly explained.**
> This MoG task was framed as a mode search problem, **meaning that success corresponds to reaching the mode that contains the goal point**. In the implementation, we compute the closest mode for each particle and considering it successful if that mode matches the goal mode.

---

> ### Comment · Reviewer_jBRb · 2025-08-03
>
> Thanks for the detailed response. However, my two concerns still remain.
>
> > This is another very intuitive question. One of the strengths of ABCD is that it can handle all these cases. This is because it sends the particles across all temperature levels. In the scenario you mentioned, ABCD would likely choose the particle sent to the highest temperature $T$, allowing ABCD to effectively perform a full restart when necessary.
>
> But doing so requires J times more compute, no? In your experiment, in Maze Large, this would mean ABCD is 32x more expensive than BoN. In real-world cases where we can't indefinitely increase the parallel compute, it could mean ABCD is at most 32x slower.
>
> Overall, the core idea of this paper is that we can mitigate the exploration-exploitation dilemma by just doing everything simultaneously. This is only possible if we can afford the additional computational cost to do so. This is the most important limitation and should be clarified explicitly.
>
> > To account for this, our ABCD framework always includes the lowest go-back temperature (i.e., 0) across all tasks.
>
> With that choice, your adaptive rule is equivalent to the naive algorithm I mentioned above, which simply "checks whether the reward has stagnated." Terminating a search algorithm when a criterion stops improving is a general idea and not particularly new. The current explanation seems to be an overly complicated way to describe this simple algorithm. As this is presented as a core contribution of the paper, upon the author's clarification, I'm lowering my score, but I'm still inclined toward acceptance and happy to be corrected.

---

> ### Author Response · Authors · 2025-08-04
>
> Thank you for your constructive feedback! We find it helpful in clarifying our paper. Please feel free to let us know if you have more questions after reading our answer in the following.
>
> > **But doing so requires J times more compute, no?** In your experiment, in Maze Large, this would mean ABCD is 32x more expensive than BoN.
>
> **No, having large J (e.g., 32) in ABCD doesn’t mean it requires 32 times more compute than BoN.** It’s important to note that ABCD’s computation depends both J and K. In our Large Maze experiment, to consider the fact that J is large, e.g., 32, we set K=1 when J=32. Thus, only a maximum of N=KxJ=32 particles are used in ABCD’s noising and denoising cycles. Therefore, a fair comparison to BoN in terms of particle count is to set N=32 for BoN.
>
> Given this, the most accurate way to compare computational efficiency is through NFE and wall-clock time rather than particle count. **For both, our experiment results clearly demonstrate that ABCD delivers substantially better performance per unit of computation.**  Specifically, as shown in the table above, ABCD with N=32 requires only 6360 NFEs to achieve a 100% success rate. In contrast, **BoN with N=32 uses 6.4 times more NFEs** (40960) yet achieves only a 94% success rate. Similarly, BoN with N=4 requires comparable NFEs (5120) to ABCD (N=32) but achieves only a 70% success rate. This same trend appears in the wall-clock time experiment which, unlike NFE, can also consider the efficiency by parallelism. ABCD with N=32 (K=1,J=32) requires only 10.5 seconds to achieve a 100% success rate, while **BoN (N=32) achieves just 94% success despite taking 4 times more time** (41.7 seconds).
> |Method|Config|Performance|Wall Clock Time(s)|NFEs|
> |-|-|-|-|-|
> |**Best-of-N (BoN)**|N=32|94±9|41.74|40960|
> |**ABCD (Ours)**|N=32 (JxK)|100±0|10.52|6360|
>
> >Your adaptive rule is equivalent to the naive algorithm which simply "checks whether the reward has stagnated." Terminating a search algorithm when a criterion stops improving is a general idea and not particularly new.
>
> We believe that the presented Adaptive Thinking Time section presents a nontrivial contribution for the following reasons.
>
> First, as we introduce the new concept of cyclic iterative refinement, we identify that **determining when to stop the inference process emerges as a new important research challenge in diffusion-based inference-time scaling**. This identification not only clarifies a new axis of difficulty in this domain but also lays the groundwork for future research to develop improved solutions. Note that, to our best knowledge, our paper is the first in this line of research to address this aspect.
>
> Second, from a technical perspective, we emphasize that we are in an interesting and distinctive setting where we can find a stopping criteria w.r.t. a distribution of particles. We introduce the new perspective that we can see from which temperature the particles came from as the uncertainty-aware criterion. This probabilistic, uncertainty-aware perspective offers a more general framework than conventional heuristics such as “stop if improvement stagnates” because this framework provides the opportunity to define different stoping criterion (e.g., measuring KL divergence between a target distribution and the current particle distribution.). In addition, in the rebuttal to Reviewer WZvc (Weakness 1), we also provide a new theoretic analysis result to show this process’s convergence property.
>
> Then, as a practical implementation of this general principle, we propose the specific stopping rule counting temperature=0 particles. While this final implementation appears similar to a familiar heuristic, it should not obscure the broader conceptual contribution: **(i) identifying a new challenge, (ii) proposing a general probabilistic perspective grounded in particle uncertainty, and (iii) demonstrating a concrete method that implements this principle**. We argue that these complete intellectual trajectory constitutes a meaningful contribution.
>
> ---
> ### **About Lowering the Score**
>
> We thank the reviewer for the constructive feedback, which greatly helped us identify key areas where our manuscript could be improved further. In addition to the clarifications above, we would like to highlight that this rebuttal also addresses **two primary concerns** raised in the review. Specifically, we have (i) added additional experimental results on **high-dimensional image generation tasks**, where our method consistently outperforms the baselines, and (ii) **included evaluations in terms of NFE alongside wall-clock time**, both of which consistently demonstrate the superiority of ABCD.
>
> We would like to ask respectfully the reviewer to consider these improvements as well when reconsidering the score.

---

### Official Review · Reviewer_Zz6X · 2025-07-01

**Clarity:** 3
**Significance:** 3
**Originality:** 3
**Rating:** 5
**Confidence:** 4

**Summary:**

This work introduces Adaptive Bi-directional Cyclic Diffusion (ABCD), a novel inference framework for diffusion models that automatically discovers improved denoising trajectories based on a given reward function. The core innovation lies in replacing fixed denoising schedules with a flexible, search-based approach that cycles bidirectionally through the diffusion process. ABCD has three main components: (1) Cyclic Diffusion Search, which alternates between denoising and re-noising steps; (2) Automatic Exploration-Exploitation Balancing, which distributes particles across multiple noise levels; and (3) Adaptive Thinking Time, which provides a stopping criterion. The authors evaluate their method on a range of tasks, including maze navigation, Sudoku solving, molecular generation, and planning, demonstrating improved performance and computational efficiency compared to baselines.

**Questions:**

- How does ABCD perform when DDIM initialization produces only poor samples? Are there any other possible failure modes?

 - Have you tested ABCD on more complex problems (such as small scale image generation)? Even modest increases in performance would provide valuable insights into scalability toward real world applications.

- What time-step schedulers and sampling algorithms (e.g. DDIM, DDPM) were used for each baseline method?

**Ethical Concerns:**

["NO or VERY MINOR ethics concerns only"]

**Final Justification:**

This paper presents an interesting and novel inference framework for diffusion models that automatically discovers improved denoising trajectories based on a specified reward function. Considering the thorough evaluation and the additional experiments provided by the authors in the rebuttal, I would like to recommend acceptance of this work.

**Limitations:**

yes

**Paper Formatting Concerns:**

The paper does not have any formatting concerns.

**Quality:**

3

**Strengths And Weaknesses:**

### **Strengths**

* The paper addresses an important limitation in current diffusion model inference—namely, the rigidity of fixed denoising schedules.
* The bi-directional cycling approach and adaptive termination are novel contributions.
* The adaptive termination criterion based on the temperature distribution is conceptually interesting.
* The writing is clear, with effective use of figures, and the method is presented with sufficient detail for reproducibility.
* The experimental evaluation spans multiple domains, demonstrating consistent improvements.
* The method is compatible with both differentiable and non-differentiable reward functions.


### **Weaknesses**

* The main weakness of the paper is the lack of experiments on more complex, real-world problems such as image generation or language modeling. Including such experiments would significantly strengthen the contribution by demonstrating ABCD’s applicability to practical scenarios. At present, it remains unclear how well the method translates to real-world tasks like text-to-image generation.

* The highest-performing configurations of ABCD still require significantly more inference time than baseline sampling. A more detailed analysis of computational overhead—including how varying the budget allocated to each ABCD component affects final quality—would be helpful for understanding the trade-offs.

* The method’s reliance on DDIM initialization may introduce failure modes that are not fully addressed. An ablation study evaluating the impact of different initialization strategies would help assess the robustness of ABCD.

---

> ### Author Rebuttal · Authors · 2025-07-31
>
> We sincerely appreciate your thoughtful feedback, including your comments on **applying our method to more complex problems, handling poor initialization, and experimental details**. Below, we provide responses to all of the questions and concerns you raised:
>
> ### **[Question 1 & Weakness 3] How does ABCD perform when DDIM initialization produces only poor samples? Are there any other possible failure modes?**
>
> Thank you for highlighting this important possible failure case. Rather than being a weakness, **we believe that possible poor initialization is actually a setting in which ABCD demonstrates its strengths**. Specifically, ABCD is particularly advantageous in scenarios with bad initializations (i.e., an initial low-quality generation), as it does not terminate after a single pass but **instead enables adaptive, iterative refinement for each sample**.
>
> Additionally, **the key hypothesis behind ABCD's design is that reaching the correct mode doesn't require high-quality fine-grained moves initially**. Instead, it's more efficient to quickly approach the good mode (by DDIM) by trading off some quality. Once we're near the appropriate mode, we can then focus on taking fine-grained moves (low-temperature moves) to refine the result.
>
> ### **[Question 2 & Weakness 1] Have you tested ABCD on more complex problems (such as small scale image generation)?**
>
> We appreciate your point and, in response, **have conducted additional experiments in more complex problems such as image generation**. **The results were consistent with our other experiments.** Please refer to our response to **[Question 1] from Reviewer WZvc** above for more details.
>
> ### **[Question 3] What time-step schedulers and sampling algorithms (e.g. DDIM, DDPM) were used for each baseline method?**
>
> Within each task, we used the same pretrained model during inference and applied the same DDIM framework across all baseline methods. However, ABCD uniquely employed a jumpy denoising schedule, where the model skips several diffusion steps during each iteration to enable faster exploration. Specifically, we used 10-step jumpy denoising over $T=50$ for Sudoku, 5-step jumpy denoising over $T=50$ for Pixel Maze, 10-step jumpy denoising over $T=256$ for OGBench, and 10-step jumpy denoising over $T=1000$ for Molecular generation.
>
> We used the jumpy denoising only in our method because that is the main inference-scaling mechanism implementing the main hypothesis of our method (described in the above answer to **[Question 1 & Weakness 3]**). For the baselines, this coarse jumpy denoising is actually harmful because, unlike ABCD, they do not allow additional chance to improve once it reaches at $x_0$.
>
> ### **[Weakness 2] A more detailed analysis of computational overhead—including how varying the budget allocated to each ABCD component affects final quality—would be helpful for understanding the trade-offs.**
>
> Thank you for raising this important point regarding computational trade-offs. **To better understand how different computational budgets affect final performance, we added ablation study analyzing the two core components of ABCD—Adaptive Thinking Time** and **Automatic Exploration–Exploitation Balancing—under varying configurations.** Specifically, we conducted experiments that vary the level of computation allocated to each mechanism to examine how efficiency and accuracy are impacted in both the Sudoku and text-to-image generation domains.
>
> In the **Sudoku task**, we focused on the **Adaptive Thinking Time mechanism** by varying the **termination percentage** and **the value of $\kappa$**, two key hyperparameters that define ABCD’s adaptive terminal condition. As shown in the two tables, **increasing $\kappa$ generally improves accuracy but also leads to longer inference time, and similarly, raising the termination percentage improves performance at the cost of additional computation**. These results show that by adjusting the strictness of the termination criterion, we can effectively control the trade-off between accuracy and efficiency.
>
> - **ABCD – Sudoku (Changing 𝜅)**
> |Metric|perc=1.0,𝜅=1|perc=1.0,𝜅=5|perc=1.0,𝜅=10|
> |------|-------------|-------------|--------------|
> |Performance|0.958|0.986|0.994|
> |Time (s)|0.443|0.608|0.754|
> |NFEs|2296|3205|3942|
> - **ABCD – Sudoku (Changing percentage)**
> |Metric|perc=0.4,𝜅=1|perc=0.6,𝜅=1|perc=0.8,𝜅=1|
> |------|-------------|-------------|-------------|
> |Performance|0.698|0.821|0.903|
> |Time (s)|0.112|0.210|0.329|
> |NFEs|520|1059|1667|
>
> In the **text-to-image generation task**, we analyzed the **Automatic Exploration–Exploitation Balancing mechanism by varying the granularity of the temperature pool**—specifically, the number of go-back temperatures included in the pool. We fixed the number of particles and terminal conditions (max_iter=50, $\kappa=30$) and varied the temperature pool size. **As shown in the table, using a finer pool (i.e., more diverse temperature values) improves the final compressibility and aesthetic score but also increases both inference time and NFE.** This experiment highlights the impact of temperature diversity on controllability and performance.
>
> - **Compressibility**
> |Temperature pool size|Compressibility|Time (s)|NFEs|
> |---|---|---|---|
> |2|-55.19±4.93|233.33|8188|
> |3|-48.84±2.14|285.83|10500|
> |6|-46.14±2.16|356.5|12800|
> - **Image Aesthetic**
> |Temperature pool size|Aesthetic|Time(s)|NFEs|
> |---|---|---|---|
> |2|6.2345±0.0535|237.92|8200|
> |3|6.3179±0.0488|286.58|10369|
> |6|6.3624±0.1419|354.67|12578|
>
> As you suggested, we will include a more comprehensive analysis of these trade-offs in the final camera-ready version. We also plan to report results using **Number of Function Evaluations (NFE)** as the x-axis instead of inference wall-clock time, as this metric provides a more standardized basis for comparing computational efficiency. A summary table is already available in **Reviewer jBRb’s [Question 4 & Weakness 5].**

---

> > ### Comment · Reviewer_Zz6X · 2025-08-04
> >
> > I would like to thank the authors for their rebuttal. My main concern regarding the application of ABCD to more complex problems has been addressed, and I will increase my score accordingly. The computational overhead analysis is also an interesting aspect of ABCD, as it helps clarify how different components contribute to the overall runtime of the method.

---

### Official Review · Reviewer_WZvc · 2025-07-02

**Clarity:** 3
**Significance:** 3
**Originality:** 3
**Rating:** 5
**Confidence:** 3

**Summary:**

The paper proposes adaptive bidirectional Cyclic Diffusion (ABCD), a search-based inference framework for adaptive inference-time scaling for diffusion models with respect to the available computation budget. The authors identified fixed denoising schedules as a bottleneck for inference time scaling, and proposed a renoising and denoising schedule. Experiment results show good performance on a variety of tasks including Sudoku, Pixel Maze, Molecule Generation and OGBench Point Maze

**Questions:**

Would the method work well for tasks such as text-to-image or text-to-video that diffusion models are widely applied to?

**Ethical Concerns:**

["NO or VERY MINOR ethics concerns only"]

**Quality:**

3

**Strengths And Weaknesses:**

(+) the idea of Adaptive Inference-Time Scaling and the ABCD frame, to my knowledge, is novel and good first explorations to this direction and can inspire future work

(+) Experiments were conducted in a comprehensive set of tasks including Sudoku, Pixel Maze, Molecule Generation and OGBench Point Maze and all show promising results

(+) ablations seems good to me

(-) The experiments and proposed method are mostly experimental instead of theoretical. Not much theoretical justification

---

> ### Author Rebuttal · Authors · 2025-07-31
>
> Thank you for your valuable comments, including your concerns regarding **the lack of theoretical justification and questions about the applicability of our method to domains such as text-to-image or text-to-video generation**. We sincerely appreciate your thoughtful feedback, and we have addressed all of your questions in detail below:
> ### **[Weakness 1] the lack of theoretical justification:**
> We thank the reviewer for comment regarding the theoretical justification of our method. We agree that a formal analysis complements our empirical results and strengthens the paper's contributions.
>
> Motivated by the reviewer's valuable feedback, **we have developed a theoretical analysis for the convergence of our automatic termination condition.** Due to the space limit, we provide a brief sketch below but the full proof will be included in the revised manuscript.
>
> ### **I. Formal statement**
> Let T = {t₁, …, tₘ}  be the “temperature pool” with lowest temperature t₁ = 0.
> At cycle c, let Dₖ(c) be the multiset of origin‐temperatures of the top‑k particles after denoising.
> **Stopping rule:** terminate at the first cycle N such that for all i in {N − κ + 1, …, N − 1, N},  Dₖ(i) = {0, 0, …, 0},  i.e., all top‑k particles originate from t₁ = 0 for κ consecutive cycles.
>
> ---
> **Assumptions**
>
> 1. **Bounded reward.** There is a reward function r: ℝᵈ → ℝ with finite supremum r* < ∞.
> 2. **Monotonic selection.** Let rₖ(c) = min{rewards of top‑k at cycle c}. Then rₖ(c) is non‐decreasing in c.
> 3. **Attainment.** There exists at least one state x* such that r(x*) = r*.
>
> ---
> **Theorem (Termination guarantee).** Under these assumptions, ABCD’s Adaptive Thinking Time criterion triggers termination in finite time.
>
> ---
> The proof of this theorem is built upon several key lemmas, which we now introduce. We will include the proof for the lemmas in the revised manuscript.
>
> **Lemma 1 (Uniform Hit‐Chance).** Let x* be any maximizer of the reward, r(x*) = r*. Under the ABCD dynamics (Algorithm 1), there is a constant p > 0 such that for each cycle c, Pr(E_c | 𝓕_{c−1}) ≥ p, where E_c = “x* appears among the selected top‑K at cycle c” and 𝓕_{c−1} is the sigma‑algebra of all randomness up to (but not including) cycle c.
>
> **Lemma 2 (Optimality).**  Under the assumptions:
>
> 1. *Monotonicity and Boundedness:* rₖ(c) is non‑decreasing and rₖ(c) ≤ r* < ∞, so rₖ(c) → r_{∞} ≤ r*.
> 2. *Supremum Attained:* there exists x* with r(x*) = r*.
> 3. *Uniform Hit‑Chance:* at each cycle c, Pr(E_c | past) ≥ p > 0.
> Then Pr[r_{∞} = r*] = 1.
>
> **Lemma 3 (Finite‐Time Hitting).** Under the *Uniform Hit‐Chance* lemma, define  T = inf{ c ≥ 1 | x* appears among the top‑K at cycle c }.  Then Pr(T < ∞) = 1, Pr(T > n) ≤ (1 − p)ⁿ  for all n ∈ ℕ, where p > 0 is the per‑cycle success lower bound.
>
> ---
> ### **II. Proof of Main Theorem**
>
> We now assemble the results from the lemmas to prove the termination guarantee theorem.
>
> 1. **Monotone convergence of rₖ(c).** The sequence rₖ(0), rₖ(1), rₖ(2), … is non‑decreasing and bounded above by r*, hence it converges to some limit r_{∞} ≤ r*.
> 2. **Limit must equal** r*. According to the Optimality Lemma, r_{∞} = r*.
> 3. **Optimal is reached in finite time.** By the Finite‑Time Hitting Lemma, rₖ(c) reaches r* at some finite cycle C.
> 4. **Once optimal is reached, only t₁‑origins survive.** At cycle C where rₖ(C) = r*, any particle noised at temperatures t > 0 and then denoised back cannot exceed r*. Thus, the only way to preserve the top‑k set of reward‑r* particles is via replicas from t₁ = 0, which leave them unchanged. Consequently, for all c ≥ C, Dₖ(c) = {0, 0, …, 0}.
> 5. **κ‑persistence triggers termination.** Therefore, once cycle C is reached, the condition “all top‑k from temperature 0 for κ consecutive cycles” is satisfied by cycle C + κ − 1, and ABCD terminates.
>
> ### **[Question 1] Would it work well for tasks such as text-to-image or text-to-video that diffusion models are widely applied to?**
> As suggested by multiple reviewers, **we evaluated ABCD on the text-to-image generation tasks using Stable Diffusion v1.5, a latent diffusion model, to generate 512×512 images $x \sim p_\theta(x|y)$ conditioned on textual prompts $y$**. **The results confirm ABCD consistently outperforms key baselines (BoN and SoP)**, exhibiting performance trends aligned with our previous experiments.
>
> For example, **with around 10k NFEs, ABCD achieves compressibility of −49.17, while SoP and BoN achieves −72.64 and −81.89, respectively. The same trend is observed w.r.t. aesthetic and human preference score.** This demonstrates ABCD’s capability to efficiently navigate the high-dimensional image space.
>
> Specifically, we comprehensively assessed ABCD’s performance using three metrics: **compressibility**, measured as negative JPEG file size (in kilobytes) after compression following [3]; **aesthetic evaluation**, computed using LAION’s V2 Aesthetic Predictor [1], a linear MLP built upon CLIP embeddings, trained on over 400,000 human ratings; and **human preference evaluation**, based on the HPSv2 scorer [2], a CLIP model fine-tuned on 798,090 human-selected rankings across 433,760 image pairs. For compressibility and aesthetic evaluations, we used animal category prompts from [3] (e.g., Dog, Cat, Panda), whereas for human preference evaluation we used human instruction prompts from [2]. Detailed quantitative results are provided below, and we will include visualizations of generated images in the camera-ready version.
> - **Compressibility**
> |Inference Method|Add.Compute|Compressibility|Clock Time(s)|NFEs|
> |----------------|:---------:|:-------------:|------------:|---:|
> |**Base Diffusion**|No add|−100.07±2.88|58.42|2000|
> |**Best-of-N(BoN)**|N=5|−81.89±5.71|257.58|10000|
> ||N=10|−77.86±8.42|494.33|20000|
> |**Search-over-Path(SoP)**|Δf=510,Δb=255|−88.36±4.88|188.75|7548|
> ||Δf=280,Δb=140|−72.64±5.61|257.67|10266|
> ||Δf=180,Δb=90|−64.59±3.43|280.25|10974|
> |**ABCD**|max_iter=30|−62.33±4.33|103.67|3340|
> ||max_iter=80|−51.49±4.34|254.50|8240|
> ||max_iter=100|**−49.17±4.29**|306.08|10176|
> - **Image Aesthetic**
> |Inference Method|Add.Compute|Image Aesthetic|Clock time (sec.)|NFEs|
> |----------------|-----------|---------------|-----------------|----|
> |**Base Diffusion**|No add|5.57 ± 0.03|58.83|2000|
> |**Best-of-N (BoN)**|N=5|5.79 ± 0.05|258.33|10000|
> ||N=10|5.86 ± 0.05|494.25|20000|
> |**Search-over-Path (SoP)**|Δf=510,Δb=255|5.68 ± 0.03|189.42|7548|
> ||Δf=280,Δb=140|5.84 ± 0.08|258.67|10266|
> ||Δf=170,Δb=85|5.87 ± 0.04|285.08|10704|
> |**ABCD**|max_iter=30|6.13 ± 0.09|105.42|3340|
> ||max_iter=80|6.23 ± 0.07|233.83|7677|
> ||max_iter=100|**6.25 ± 0.09**|288.42|9147|
> - **Human Preference Score**
> |Inference Method|Add.Compute|Human Preference|Clock time (sec.)|NFEs|
> |----------------|-----------|----------------|-----------------|----|
> |**Base Diffusion**|No add|0.2710 ± 0.0028|61.00|2000|
> |**Best-of-N (BoN)**|N=5|0.2753 ± 0.0017|260.92|10000|
> |**Search-over-Path (SoP)**|Δf=510,Δb=255|0.2725 ± 0.0031|192.83|7548|
> ||Δf=230,Δb=115|0.2751 ± 0.0029|272.25|10218|
> ||Δf=130,Δb=65|0.2765 ± 0.0046|811.25|11232|
> |**ABCD**|max_iter=30|0.2797 ± 0.0023|115.92|3340|
> ||max_iter=80|0.2815 ± 0.0030|272.08|8060|
> ||max_iter=100|**0.2819 ± 0.0032**|322.08|9718|
>
> [1] Christoph Schuhmann and Romain Beaumont. Laion-aesthetics. LAION. AI, 2022.
>
> [2] Wu, Xiaoshi, et al. Human preference score v2: A solid benchmark for evaluating human preferences of text-to-image synthesis. (2023).
>
> [3] Black, Kevin, et al. Training diffusion models with reinforcement learning.(2023).

---

### Official Review · Reviewer_D4Gw · 2025-07-02

**Clarity:** 4
**Significance:** 3
**Originality:** 3
**Rating:** 5
**Confidence:** 3

**Summary:**

The paper tackles the problem of inference-time optimization in diffusion models when a predefined reward function is available. The authors propose Adaptive Bi-directional Cyclic Diffusion (ABCD), a search-based inference framework composed of three main steps.
First is the Cyclic Diffusion Search: $N$ particles are denoised from pure Gaussian noise, evaluated using the reward function, and the top-$K$ are selected. These are then re-noised, and the denoise-rank-renoise cycle is repeated. To adaptively balance exploration and exploitation, the method re-noises to multiple noise levels rather than a fixed one, with higher noise levels corresponding to more exploratory search and the lower noise levels corresponding to local refinement. The cycle terminates once all top-$K$ samples are obtained from low renoised samples, indicating local convergence.
The authors evaluate ABCD on several reasoning and planning tasks, including Sudoku and maze solving, showing impressive  improvements over previous methods.

**Questions:**

* How many denoising steps are used at each of the noise levels $\{t_1, t_2, \dots, t_M\}$? Is this a constant fixed amount or does it change based on the noise level?
* Does the algorithm utilize any form of "history" to save the best top-K during the different cycles? This question mainly comes from the fact that the curves in Figures 6,7 are not always increasing w.r.t. the inference time which seems counter-intuitive.

**Ethical Concerns:**

["NO or VERY MINOR ethics concerns only"]

**Final Justification:**

The paper proposes an elegant test-time scaling approach for improving diffusion model sampling and evaluate their approach on several puzzle/reasoning/maze benchmarks with impressive results. I recommend accepting this paper.

**Limitations:**

Yes.

**Paper Formatting Concerns:**

No concerns.

**Quality:**

4

**Strengths And Weaknesses:**

**Strengths:**
* The paper is very well written and easy to follow.
* The proposed approach is simple, intuitive, and quite elegant.
* The experimental improvements are impressive, with the proposed approach improving on both speed and quality (success rate).

**Weaknesses:**
The paper is an extremely good one, and the following weaknesses are more nitpicks rather than actual weaknesses.
* All experiments are done on relatively low-dimensional data. As pointed out by the authors, it remains unclear how well the approach would scale to high-dimensional outputs, such as high-resolution text-to-image generation.
* One of the main concepts of the paper is using multiple noise levels during the renoising step, discussed in 3.2. However, denoising higher noise levels requires more function evaluations of the base model than lower noise levels. As a result, during the end of the search process, where only local refinement is needed, some amount of unnecessary compute is being performed by distributing each sample uniformly along the $M$ noise levels.

---

> ### Author Rebuttal · Authors · 2025-07-31
>
> Thank you for your constructive feedback, including your comments on **applying our method to high-dimensional outputs, concerns regarding unnecessary computation, and detailed questions about our experimental setup**. We sincerely appreciate your thoughtful observations, and we have provided responses to all of your questions and concerns below:
>
> ### **[Weakness 1] It is unclear how well your method would scale to high-dimensional outputs, such as high-resolution text-to-image generation?**
> We appreciate your point and, in response, **have conducted additional experiments in a high-dimensional domain. The results were consistent with other experiments, outperforming the rest of the baselines**. For more details, please refer to our response to **[Question 1] from Reviewer WZvc** below.
> ### **[Weakness 2] Some amount of unnecessary compute is being performed by distributing each sample uniformly along the noise levels:**
> Thank you for highlighting this insightful point—it aligns closely with our own considerations during the development of ABCD. Indeed, **we explored a more intelligent go-back strategy but found it doesn’t show meaningful improvement.**
>
> Specifically, we tested strategy where particles were redistributed to different noise levels in proportion to the average reward observed at those levels across the population. **However, as shown in the table below, despite its intuitive appeal, the performance improvement was marginal compared to the added complexity.** Therefore, we ultimately chose a simpler and more efficient formulation for the current version of ABCD. Nevertheless, as you rightly noted, we believe that incorporating smarter strategies—such as applying ideas from SMC to the temperature pool—could be a promising direction.
>
> - **Adaptive ABCD vs. ABCD in Sudoku**
> |Method|Add. Compute|Accuracy|Wall Clock Time (sec.)|
> |-------|----------|----------|---------------------|
> |**Adaptive ABCD**|perc=0.6,𝜅=1|0.810|0.220|
> ||perc=0.8,𝜅=1|0.895|0.395|
> ||perc=1.0,𝜅=1|0.951|0.489|
> ||perc=1.0,𝜅=3|0.979|0.549|
> |**ABCD**|perc=0.6,𝜅=1|0.821|0.240|
> ||perc=0.8,𝜅=1|0.903|0.350|
> ||perc=1.0,𝜅=1|0.958|0.450|
> ||perc=1.0,𝜅=3|0.979|0.557|
> ### **[Question 1] How many denoising steps are used at each of the noise levels ${t_1, t_2, ..., t_M}$? Is this a constant fixed amount or does it change based on the noise level?**
> **The denoising (DDIM) step size is fixed to the same constant across all temperatures (noise levels).** For example, for sudoku task, the step size is set to 10. So, the actual number of denoising steps are set to remaining_total_steps(t_k)/ddim_step_size.
>
> ### **[Question 2] Does the algorithm utilize any form of "history" to save the best top-K during the different cycles? This question mainly comes from the fact that the curves in Figures 6,7 are not always increasing w.r.t. the inference time which seems counter-intuitive:**
> **Yes, ABCD utilizes a form of history to save the best top-K during the cycles.** More specifically, we included temperature=0 in the go-back temperature pool, allowing the algorithm to preserve the best particles from the previous cycle unless further improvements are made. This allows the algorithm to keep the best case and thus improve monotonically.
>
> Due to this mechanism, ABCD's performance in terms of verifier score consistently improves over time. **In contrast, some baselines in Figure 6 lack this property and occasionally show performance decreases.**
>
> However, in Figure 7, we observe some performance drops of ABCD mainly in panels (a), (b), and (c). For panels (a) and (b), this occurs because we intentionally excluded the memory feature that maintains the temperature=0 configuration as part of our ablation study. For panel (c), the slight drop in success rate results from a mismatch between the verifier score (which determines our top-K selection) and the actual success rate, which aren't perfectly correlated. We will clarify this in the revision.

---

> > ### Comment · Reviewer_D4Gw · 2025-08-06
> >
> > I thank the authors for their response. I appreciate the additional experiments, and am glad to see ABCD being successful in  the high-dimensional text-to-image setting. I'm curious how much diversity is lost in the text-to-image experiments due to the repeated top-K filtering. Do all of the final images look roughly similar (i.e. converge to the same mode of the distribution)? If so, a discussion on this would be greatly welcome in the final version of the paper.
> >
> > In any case, my questions have been answered and I keep my rating and recommend accepting this paper.

---

> > > ### Author Response · Authors · 2025-08-07
> > >
> > > Thank you for raising this important point about diversity preservation. We believe ABCD effectively maintains diversity in the generated images through its core mechanism.
> > >
> > > ABCD selects top-k candidates and duplicates them across different temperatures, enabling restarts from multiple noise levels. **This approach helps preserve diversity by creating opportunities to explore distinct modes of the distribution, rather than converging to a single one**.
> > >
> > > **Following your suggestion, we evaluated the diversity of the generated images and found that ABCD does not exhibit mode collapse**. While optimization naturally reduces some diversity, our method successfully preserves the underlying multimodality of the pretrained model while steering toward high-reward regions.
> > >
> > > Specifically, we measured diversity using the average pairwise cosine similarity of CLIP embeddings, where a lower score reflects greater variability in outputs and broader exploration of the data space. Our results show that ABCD achieves an effective balance between diversity and reward, maintaining substantial output variety while significantly enhancing reward values. Visualizations of the generated images for each class and prompt, which also demonstrate this diversity, will be included in the camera-ready version. We appreciate your thoughtful comment, which has helped strengthen the paper.
> > >
> > > | Aesthetic       | Add compute      | Diversity         |
> > > |-----------------|------------------|-------------------|
> > > | Base Diffusion  | –                | 0.2957 ± 0.0481   |
> > > | BoN             | N=5              | 0.2870 ± 0.0102   |
> > > |                 | N=10             | 0.2919 ± 0.0285   |
> > > | SoP             | Δf=510, Δb=255   | 0.2912 ± 0.0308   |
> > > |                 | Δf=280, Δb=140   | 0.3005 ± 0.0199   |
> > > |                 | Δf=170, Δb=85    | 0.2672 ± 0.0323   |
> > > | ABCD            | max_iter=30      | 0.2848 ± 0.0221   |
> > > |                 | max_iter=80      | 0.2888 ± 0.0222 |
> > > |                 | max_iter=100     | 0.2880 ± 0.0242   |
> > >
> > > | Compressibility | Add compute      | Diversity         |
> > > |-----------------|------------------|-------------------|
> > > | Base Diffusion  | –                | 0.2957 ± 0.0481   |
> > > | BoN             | N=5              | 0.2859 ± 0.0190   |
> > > |                 | N=10             | 0.2781 ± 0.0184   |
> > > | SoP             | Δf=510, Δb=255   | 0.2795 ± 0.0375   |
> > > |                 | Δf=280, Δb=140   | 0.2975 ± 0.0512   |
> > > |                 | Δf=180, Δb=90    | 0.2798 ± 0.0154   |
> > > | ABCD            | max_iter=30      | 0.3128 ± 0.0084   |
> > > |                 | max_iter=80      | 0.3162 ± 0.0028 |
> > > |                 | max_iter=100     | 0.3037 ± 0.0171   |

---

### Decision · Program_Chairs · 2025-09-17

**Decision:**

Accept (poster)

**Comment:**

This paper introduces Adaptive Bi-directional Cyclic Diffusion to address the challenge of adaptive inference-time scaling. Reviewers are satisfied with the writing of this paper and impressive experimental results of the proposed method. The main concerns, including low-dimensional data and detailed comparison with other methods, are addressed with further experimental results provided by the authors. In the end, the reviewers all recommend accept. With these new results added into the revised version of this submission, I think this work meets the quality threshold of NeurIPS conference and recommend acceptance.